# Hydrogenated Cs$_2$AgBiBr$_6$ for significantly improved efficiency of lead-free inorganic double perovskite solar cell

Zeyu Zhang[1], Qingde Sun[2], Yue Lu [1✉], Feng Lu[3], Xulin Mu[1], Su-Huai Wei [2✉] & Manling Sui [1✉]

Development of lead-free inorganic perovskite material, such as Cs$_2$AgBiBr$_6$, is of great importance to solve the toxicity and stability issues of traditional lead halide perovskite solar cells. However, due to a wide bandgap of Cs$_2$AgBiBr$_6$ film, its light absorption ability is largely limited and the photoelectronic conversion efficiency is normally lower than 4.23%. In this text, by using a hydrogenation method, the bandgap of Cs$_2$AgBiBr$_6$ films could be tunable from 2.18 eV to 1.64 eV. At the same time, the highest photoelectric conversion efficiency of hydrogenated Cs$_2$AgBiBr$_6$ perovskite solar cell has been improved up to 6.37% with good environmental stability. Further investigations confirmed that the interstitial doping of atomic hydrogen in Cs$_2$AgBiBr$_6$ lattice could not only adjust its valence and conduction band energy levels, but also optimize the carrier mobility and carrier lifetime. All these works provide an insightful strategy to fabricate high performance lead-free inorganic perovskite solar cells.

[1] Beijing Key Laboratory of Microstructure and Properties of Solids, Faculty of Materials and Manufacturing, Beijing University of Technology, 100124 Beijing, China. [2] Beijing Computational Science Research Center, 100193 Beijing, China. [3] Department of Electronic Science and Engineering, and Tianjin Key Laboratory of Photo-Electronic Thin Film Device and Technology, Nankai University, 300071 Tianjin, China. ✉email: luyue@bjut.edu.cn; suhuaiwei@csrc.ac.cn; mlsui@bjut.edu.cn

As one of the significant developments of third-generation solar cells, organic-inorganic halide perovskites (OIHP) have attracted tremendous attentions since their highest photoelectronic conversion efficiency (PCE) is now approaching 25.7%[1], comparable to the silicon-based solar cells. Traditionally, the OIHP materials have a stoichiometric ratio of ABX$_3$, in which the A, B and X sites are consisted of organic cation (Methylamine cation (MA$^+$) or Formamidine cation (FA$^+$)), metal cation Pb$^{2+}$ and halide anions (Cl$^-$, Br$^-$ and I$^-$), respectively[2–5]. Despite the excellent photovoltage performance of OIHP solar cells, there still exist two critical issues that hinder their widely applications in industry[6–8]. The first aspect is the intrinsic instability of the OIHP caused by the volatility of organic components, and the other is the disreputable toxicity of lead element. To solve these problems, extensive efforts have been done to explore the Pb-free inorganic perovskite materials as the core adsorbent layer in solar cell[9–13]. For example, it has been reported that the substitution of MA$^+$ with Cs$^+$ cations in A site could increase the decomposition energy of perovskite from −0.111 eV (MAPbI$_3$) to −0.069 eV (CsPbI$_3$)[14], thereby enhancing the stability. Considering on the toxicity of Pb element in B site of perovskite, tin cation (Sn$^{2+}$) or silver cation (Ag$^+$) combined with bismuth cation (Bi$^{3+}$) are alternative substitutions to relieve the toxicity of the optical absorption layer. Nevertheless, the susceptible Sn$^{2+}$ is easily oxidized to Sn$^{4+}$ in air, which rapidly decreases the photovoltage performance of CsSnI$_3$-based solar cells[15,16]. While, the double substitutions of Pb$^{2+}$ utilizing B-site cations such as Ag$^+$ and Bi$^{3+}$ in perovskite could effectively improve the stability due to its enhanced Coulomb interaction energy, which leads to a high positive decomposition energy in Cs$_2$AgBiBr$_6$ (0.38 eV)[17]. Thus, Cs$_2$AgBiBr$_6$-based perovskite solar cell (PSC) is one of the most promising candidates in inorganic lead-free perovskite photovoltaic devices.

However, until now, although abundant attempts have been done to optimize the optoelectronic properties of Cs$_2$AgBiBr$_6$ PSC, the PCE of Cs$_2$AgBiBr$_6$ PSCs increased gradually and the champion PCE of Cs$_2$AgBiBr$_6$ PSC was just 4.23%[18–23], which is much lower than that of organic-inorganic hybrid lead-based PSCs. Therefore, the improvement of PCE in Cs$_2$AgBiBr$_6$ PSC is urgently needed. There are three main factors restricting the improvement of the efficiency, including the large bandgap ($E_g$), low carrier mobility and carrier lifetime. The Cs$_2$AgBiBr$_6$ has an indirect measured bandgap of 1.83 eV–2.19 eV[24,25], which limits the light absorption of low energy photons in the perovskite layer. Meanwhile, the carrier mobility of Cs$_2$AgBiBr$_6$ is only about 1.00 cm$^2$V$^{-1}$s$^{-1}$–11.81 cm$^2$V$^{-1}$s$^{-1}$, which is quite smaller than the one in MAPbI$_3$ (about 35 cm$^2$V$^{-1}$s$^{-1}$)[26–28]. In addition, the carrier lifetime of thin-film Cs$_2$AgBiBr$_6$ is only about 13.7 ns due to a large number of defect states at the grain boundaries of the films[29,30].

In order to improve the PCE of Cs$_2$AgBiBr$_6$ PSC, extensive efforts have been tried to improve the carrier transport properties. For example, Wang et al. successfully developed a vapor-deposition method to fabricate the Cs$_2$AgBiBr$_6$ perovskite film with dense grains, whose defect density ($2.13 \times 10^{16}$ cm$^{-3}$)[18] was lower than that of the solution method ($9.1 \times 10^{16}$ cm$^{-3}$)[31]. Gao et al. spin-coated an excellent Cs$_2$AgBiBr$_6$ film with the grain size of 410 nm via a proper anti-solvent technology, the defect density of perovskite film has been decreased significantly and the PCE of Cs$_2$AgBiBr$_6$ PSCs has been improved up to 2.2%[19]. On the other side, reducing the bandgap $E_g$ of perovskite has been achieved in several reports. For example, Mete et al. found that doping of Co into PbTiO$_3$ perovskite could effectively increase the light absorption property[32]. By doping Tl$^+$, Tl$^{3+}$, Sb$^{3+}$ or Sn$^{2+}$ elements into the lattice of Cs$_2$AgBiBr$_6$, the bandgap has been found to decrease effectively[33–35]. In addition, Li et al. successfully

decreased the $E_g$ value of single crystal Cs$_2$AgBiBr$_6$ from 2.19 eV to 1.7 eV by applied a high pressure of 15 GPa, however, the crystal structure as well as the $E_g$ value recovered rapidly as the pressure is removed[36]. Recently, by adjusting the degree of Ag-Bi disorder, Ji et al. achieved the decrease of bandgap from 1.98 eV to 1.72 eV in single crystal Cs$_2$AgBiBr$_6$, which was grown at high evaporation temperature of 150 °C[37]. This result is consistent with the predication of Yang et al. based on density functional theory (DFT) calculations that the bandgap of Cs$_2$AgBiBr$_6$ could be effectively decreased from 1.93 eV to 0.44 eV after disordering at the (Ag, Bi) sublattice[38]. However, due to unsuitable pressure and temperature preparation conditions, such effective crystal-engineering approaches for the reduction of bandgap could not been implemented in the spin-coated Cs$_2$AgBiBr$_6$ thin films for the fabrication of PSC devices. Therefore, the critical issue is the feasibility of the strategy of reducing bandgap in the perovskite thin films and its realizability in PSCs.

In this text, by using a hydrogenation method, atomic hydrogen could be successfully doped into the interstitial sites of Cs$_2$AgBiBr$_6$ crystal lattice, which modified the bandgap of high-quality double perovskite Cs$_2$AgBiBr$_6$ films from 2.18 eV to 1.64 eV. Based on this, the highest PCE of hydrogenated Cs$_2$AgBiBr$_6$-based PSC has been improved for more than 150% up to 6.37%, which is the record-high efficiency so far. Moreover, the hydrogenated Cs$_2$AgBiBr$_6$ PSCs exhibit remarkable stability in nitrogen environments under light illumination at both room temperature and as high as 85 °C.

## Results

**Fabrication and characterization of hydrogenated Cs$_2$AgBiBr$_6$ films.** As shown in Fig. 1a, high quality Cs$_2$AgBiBr$_6$ films were firstly fabricated by using a traditional one-step spin-coating solution method (details see Method), which have a face-centered cubic structure phase (Fm$\overline{3}$m) (the black spectrum in Fig. 1b) and present a yellow color (the optical image of the pristine sample insets in Fig. 1b). Then the perovskite films were hydrogenated for different times (600 s and 1200 s) by using plasma treatment in a hydrogen (H$_2$) and argon (Ar) mixed gas environment (Fig. 1a, see Method). When the hydrogenation time is up to 1200 s, the Cs$_2$AgBiBr$_6$ perovskite films remain to be the cubic phase (Fm$\overline{3}$m) without any phase transition (Fig. 1b), but the color of the films change from yellow into black (insert pictures in Fig. 1b) and the surface morphology of perovskite film becomes smoother (Fig. 1c).

To quantitatively analyze the color change of the hydrogenated Cs$_2$AgBiBr$_6$ films, optical absorption of these perovskite films was investigated by Ultraviolet-visible absorption spectra (UV-vis) as shown in Fig. 1d. When increasing the hydrogenation time from 0 s to 1200 s, the light absorption for photons with wavelength above 496 nm exhibits an obvious increase, which indicates an effective improvement of visible light absorption of the hydrogenated Cs$_2$AgBiBr$_6$ perovskite films. Meanwhile, the bandgap of hydrogenated Cs$_2$AgBiBr$_6$ films is also consistent with the measured one determined by the tauc plots equation [Eq. 1],

$$(\alpha h\nu)^{1/n} = A(h\nu - E_g) \tag{1}$$

where $\alpha$ is absorption coefficient, $h$ is Plank's constant, $\nu$ is the incident frequency, $A$ is a constant of proportionality[39] and the exponent $n$ is 2 for indirect bandgap semiconductor[40]. As shown in Supplementary Fig. 1, the pristine Cs$_2$AgBiBr$_6$ film is an indirect semiconductor with a bandgap value of 2.18 eV, which is consistent with previous reports[24,25]. However, after hydrogenating the Cs$_2$AgBiBr$_6$ films for 600 s and 1200 s, the treated Cs$_2$AgBiBr$_6$ films are still indirect semiconductors, and their bandgap values have been decreased from 2.18 eV to 1.91 eV and

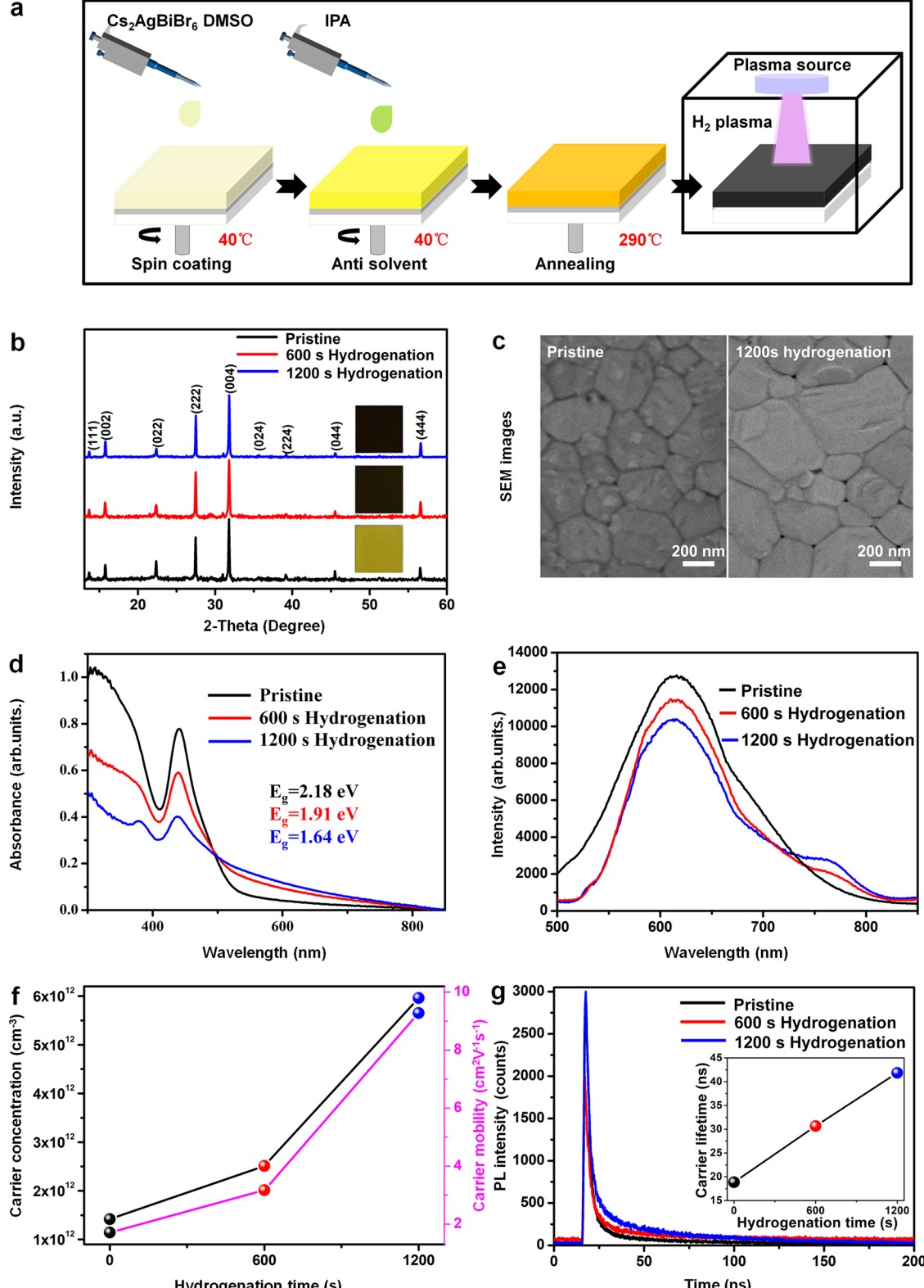

**Fig. 1 Fabrication of the hydrogenated Cs₂AgBiBr₆ perovskite films and the photoelectrical property characterization. a** The preparation method of Cs₂AgBiBr₆ perovskite films during the hydrogenation treatment in hydrogen gas plasma. **b** X-ray Diffraction (XRD) patterns of the Cs₂AgBiBr₆ perovskite films with different hydrogenation time (inserts show the optical pictures of the corresponding Cs₂AgBiBr₆ films). **c** Scanning electron microscopy (SEM) images of the pristine Cs₂AgBiBr₆ film and the 1200 s hydrogenated Cs₂AgBiBr₆ film. **d** Ultraviolet-visible absorption spectra (UV-vis) of the Cs₂AgBiBr₆ perovskite films with different hydrogenation time. **e** Photoluminescence (PL) spectra of the Cs₂AgBiBr₆ perovskite films with different hydrogenation time. **f** Carrier mobility and carrier concentration of the Cs₂AgBiBr₆ perovskite films with different hydrogenation time. **g** Time-resolved photoluminescence (TRPL) and the measurement of carrier lifetime (inset) of Cs₂AgBiBr₆ perovskite films with different hydrogenation time.

further to 1.64 eV, respectively (Fig. 1d). In addition, as the increasing of hydrogenation time, a broad photoluminescence (PL) peak at around 760 nm raises continuously (Fig. 1e), which indicates a reduction of bandgap of $Cs_2AgBiBr_6$ film during hydrogenation treatment.

It should be pointed out that some groups ascribed the enhancement of sub-bandgap adsorption to the defect state generation between the valence band (VB) and conduction band (CB), but always taking along with a reduction of PL intensity and lifetime[41–43]. In order to further understand the effect of plasma treatment on the photoelectrical properties of $Cs_2AgBiBr_6$ film, its carrier mobility, carrier concentration and carrier lifetime are tested as shown in Fig. 1f and g. The carrier mobility in the pristine $Cs_2AgBiBr_6$ film was about 1.71 $cm^2V^{-1}s^{-1}$ and obtained from the Hall-effect measurement data (Supplementary Fig. 2). After hydrogenating the $Cs_2AgBiBr_6$ film for 600 s, the carrier mobility (3.17 $cm^2V^{-1}s^{-1}$) increases by 85%. As elongating the hydrogenation time to 1200 s, the carrier mobility (9.28 $cm^2V^{-1}s^{-1}$) increases up to 542%, as compared with the one in the pristine $Cs_2AgBiBr_6$ film (Fig. 1f). Meanwhile, the carrier concentration of $Cs_2AgBiBr_6$ film has been improved from $1.42 \times 10^{12}$ $cm^{-3}$ to $2.51 \times 10^{12}$ $cm^{-3}$ and further to $5.96 \times 10^{12}$ $cm^{-3}$ after 600 s and 1200 s hydrogenation treatment (Fig. 1f), which is mostly ascribing to the enhancement of light absorption range above 496 nm. Finally, an obvious increase of the carrier lifetime from 18.85 ns to 30.67 ns and further to 41.86 ns in hydrogenated $Cs_2AgBiBr_6$ film has been detected (inset in Fig. 1g, Supplementary Fig. 3 and Supplementary Table. 1) after hydrogenation treatment of the $Cs_2AgBiBr_6$ films for 600 s and 1200 s, which is quite different from the decreasing of PL lifetime that is induced by the defect state generation for sub-bandgap adsorption[41,42]. Therefore, considering on the results in UV-vis spectra, PL spectrum and PL lifetime, the bandgap of $Cs_2AgBiBr_6$ film was indeed reduced after hydrogenation, but not induced by the increasing of defect sate density in perovskite film[41–43].

In order to explore the influence of hydrogen atom ($H^*$) on $Cs_2AgBiBr_6$, the distribution of $H^*$ in $Cs_2AgBiBr_6$ film needs to be studied. After hydrogenating the perovskite films from 0 to 3000 s, the color evolution rate of the front (surface exposed in the hydrogen plasma environment) and back (see from the glass side of the film) sides of the $Cs_2AgBiBr_6$ films is quite different (Fig. 2a). In order to quantify the variation tendency for the front and back sides of the hydrogenated $Cs_2AgBiBr_6$ films, the blackness of the optical images on both sides was counted up as a change of hydrogenation time (black and red curves in Fig. 2a). It is clearly identified that the blackness keeps increasing gradually with the hydrogenation time increase, and the blackness of front side is always higher than the one of back side, especially when hydrogenating the $Cs_2AgBiBr_6$ film for 600–1800 s. After 1800 s hydrogenation treatment, the blackness difference of front and back sides becomes smaller. All these results indicate that the $H^*$ concentration in the hydrogenated $Cs_2AgBiBr_6$ film has a gradient distribution depending on the positing and hydrogenating time.

During the hydrogenation treatment of $Cs_2AgBiBr_6$ film (Method), $H^*$ needs to diffuse firstly from the front side into the inner of perovskite film, so it obviously obeys the one-dimensional constant surface concentration diffusion model[44]

$$\frac{C}{C_s} = \mathrm{erfc}\left(\frac{x}{2\sqrt{Dt}}\right) \qquad (2)$$

where $x$ is the diffusion thickness, $D$ is diffusion coefficient of $H^*$ in $Cs_2AgBiBr_6$ film, $t$ is the diffusion time (equivalent to hydrogenation time), $C_s$ is diffusion source concentration ($x = 0$) and $C$ is concentration at the depth $x$ in film [Eq. 2]. However, due to the uncertainty of the diffusion coefficient of $H^*$ in $Cs_2AgBiBr_6$, it is hard to predict quantitatively the $H^*$

distribution concentration in perovskite film. The diffusion coefficient of $H^*$ in most materials ranges from $4.9 \times 10^{-15}$ $cm^2/s$ to $1.07 \times 10^{-5}$ $cm^2/s$ (Supplementary Table. 2)[45–52]. So here, to estimate the diffusion coefficient (detail see "Method" section), we first simulated the depth-dependent $\frac{C}{C_s}$ values at different hydrogenation time (600 s, 1200 s, 1800 s, 2400 s and 3000 s) based on the selected diffusion coefficients from $1 \times 10^{-15}$ $cm^2/s$ to $1 \times 10^{-5}$ $cm^2/s$, as shown in Supplementary Fig. 4a–e, respectively. At a maximum film thickness of 140 nm (Supplementary Fig. 4f), the simulated $\frac{C}{C_s}$ values are shown in Supplementary Fig. 4 and summarized in Supplementary Table. 3. Considering the simulated $\frac{C}{C_s}$ value at the real film thickness should be comparable to the image blackness (details see Method and Fig. 2a) ratio at the front and back sides of hydrogenated $Cs_2AgBiBr_6$ film (Supplementary Fig. 4g and Supplementary Table. 3), the real diffusion coefficient $D$ of $H^*$ in $Cs_2AgBiBr_6$ is then estimated to range from about $1 \times 10^{-14}$ $cm^2/s$ to $1 \times 10^{-13}$ $cm^2/s$. Figure 2b shows a schematic diagram of $H^*$ diffusion in the hydrogenated $Cs_2AgBiBr_6$ film of 1200 s.

After the diffusion of $H^*$ into the $Cs_2AgBiBr_6$ lattice, the perovskite films exhibit extraordinary stability in $N_2$ environment undergoing long-term high temperature (85 °C) or 1 sun light illuminating treatments (Fig. 2c and d). After 80 day ageing process, these hydrogenated $Cs_2AgBiBr_6$ films keep optically black, indicating their superior stability in $N_2$ environment with light and heat treatments. The excellent photoelectric properties of hydrogenated perovskite films mentioned above (Fig. 1) suggest that the hydrogenated $Cs_2AgBiBr_6$ should have great application potential in the lead-free PSC. In order to fabricate the PSC practically, energy level of hydrogenated $Cs_2AgBiBr_6$ needs to be firstly determined as shown in Fig. 2e. Here, the ultraviolet photoelectron spectroscopy (UPS) characterizations confirm that the work function $\phi$ of 0 s, 600 s and 1200 s hydrogenated $Cs_2AgBiBr_6$ films were −5.27 eV, −4.39 eV and −3.89 eV, and the energy gap $\Delta E$ between the energy level of maximum VB ($E_{VBM}$) and Fermi level were 0.93 eV, 1.26 eV and 1.44 eV (Supplementary Fig. 5), respectively. Combining with the bandgap values in Fig. 1d, $E_{VBM}$ and minimum CB ($E_{CBM}$) of hydrogenated $Cs_2AgBiBr_6$ films could be obtained as shown in Fig. 2e (details see Supplementary Fig. 5).

**Hydrogenated $Cs_2AgBiBr_6$ perovskite solar cells**. To make sure the compatibility of energy level in PSC, the energy level positions of $SnO_2$ and Spiro-OMeTAD were also determined through UV-vis and UPS spectra as shown in Supplementary Fig. 6, from which we could find that they matched well with the energy level of hydrogenated $Cs_2AgBiBr_6$ films (Fig. 2f). So here, we selected $SnO_2$, Spiro-OMeTAD, indium tin oxide (ITO) and gold (Au) as the electron transport layer (ETL), hole transport layer (HTL) and electrodes (Fig. 2f), respectively. And the PSC device was successfully prepared as shown in Supplementary Fig. 7, in which the layer structures could be identified clearly. The pristine $Cs_2AgBiBr_6$-based PSC exhibits a low PCE of only about 0.55%, and the short-circuit current density ($J_{sc}$) is 1.03 mA/cm² (see black lines and dots in Fig. 3a and d). After hydrogenating the $Cs_2AgBiBr_6$ films for 600 s and 1200 s, although the open-circuit voltage ($V_{oc}$) has a little increase from 0.88 V to 0.92 V, the short-circuit current density ($J_{sc}$) increases significantly from 1.03 mA/cm² to 7.65 mA/cm² and further to 11.40 mA/cm² (Fig. 3a). For the 1200 s hydrogenated $Cs_2AgBiBr_6$ PSCs, the champion PCE values are 5.64% and 6.37% for forward and backward scans, respectively (Fig. 3b). And this is the record high efficiency in the $Cs_2AgBiBr_6$-based PSC devices so far. The corresponding photocurrent and PCE at maximum power output of 0.64 V bias are

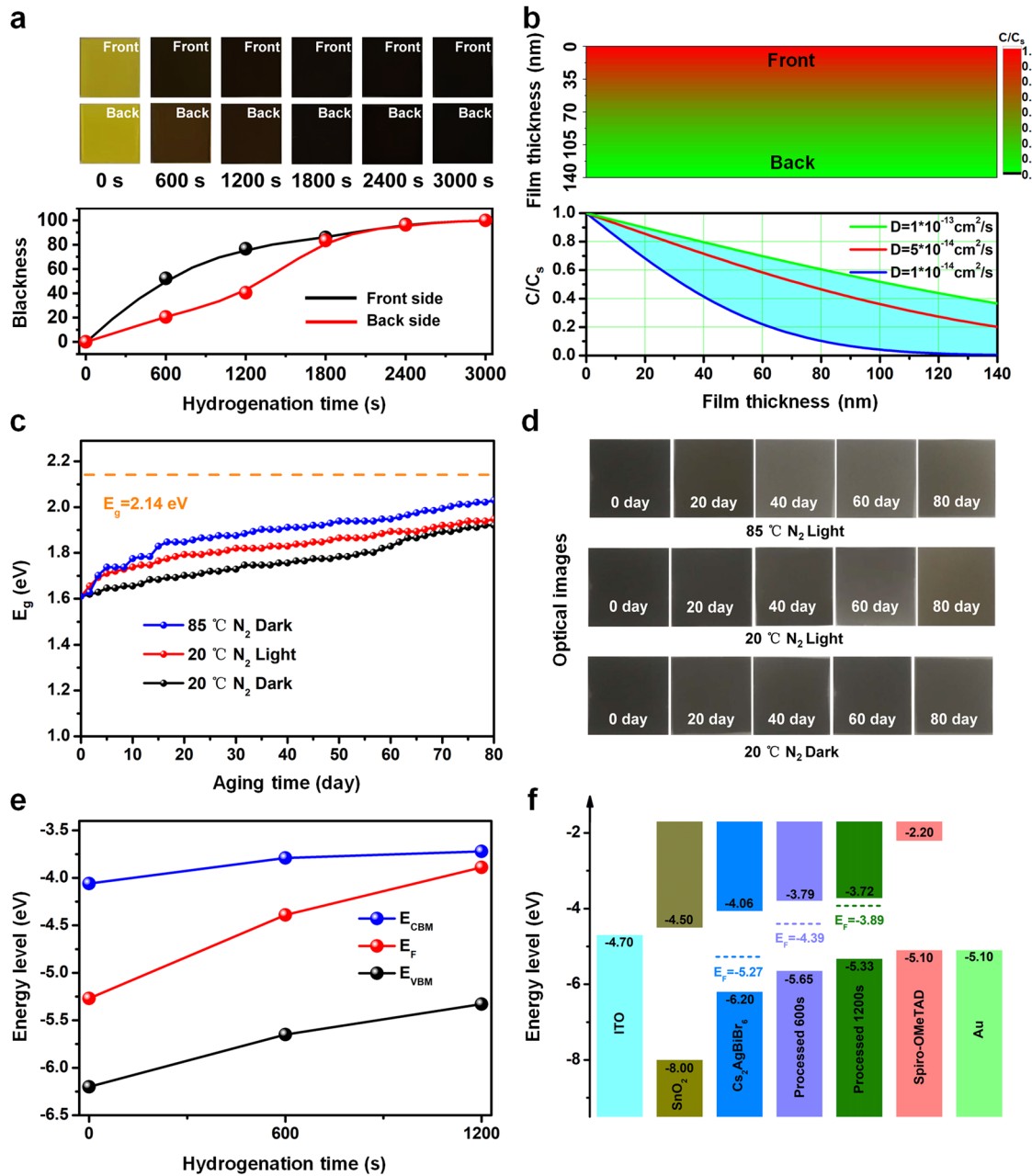

**Fig. 2 The distribution of atomic hydrogen, environmental stability and energy level in the hydrogenated Cs$_2$AgBiBr$_6$ film. a** The optical images of the Cs$_2$AgBiBr$_6$ films with different hydrogenation time (top) and the blackness evolution of their corresponding images at the front and back sides (bottom). **b** The schematic diagram of the simulated distribution of atomic hydrogen in hydrogenated Cs$_2$AgBiBr$_6$ film according to a diffusion coefficient of 5 × 10$^{-14}$ cm$^2$/s (top), and the diffusion curve of H$^*$ based on different diffusion coefficient at the depth direction of film (bottom). **c** Evolution of the bandgap values of hydrogenated Cs$_2$AgBiBr$_6$ films as storing in N$_2$, 1 sun light illumination and 85 °C heating environments. **d** The photo-images (Front side) for stability of the hydrogenated Cs$_2$AgBiBr$_6$ films at N$_2$, light illumination and 85 °C heating environments, respectively. **e** The energy level of $E_{CBM}$, $E_{VBM}$ and $E_F$ (femi level) in Cs$_2$AgBiBr$_6$ films with different hydrogenation time (0 s, 600 s and 1200 s). **f** Energy-level diagram of the functional layers in hydrogenated Cs$_2$AgBiBr$_6$ perovskite solar cell.

shown in Fig. 3c (the photocurrent of pristine and 600 s hydrogenated PSCs were shown in Supplementary Fig. 8). It could clearly observe that hydrogenated Cs$_2$AgBiBr$_6$ PSC devices exhibit a rapid response after light soaking, resulting in stable PCE (about 6.25% for 1200 s hydrogenated sample) and current density (9.78 mA/cm$^2$) during the continuous light illumination. Meanwhile, the average PCE of hydrogenated Cs$_2$AgBiBr$_6$ is effectively improved from 0.42% (0 s) to 2.95% (600 s) and then to 5.56% (1200 s) (Fig. 3d). The statistical analysis of $V_{oc}$, $J_{sc}$ and fill factor (FF) were summarized in Supplementary Fig. 9.

To further confirm the decrease of bandgap value and enhancement of $J_{sc}$, external quantum efficiency (EQE) spectra of Cs$_2$AgBiBr$_6$ PSCs with different hydrogenation time (0 s, 600 s and 1200 s) are shown in Fig. 3e. The photoresponse range expands gradually from 539 nm for pristine to 720 nm for 1200 s hydrogenated Cs$_2$AgBiBr$_6$ PSCs, and matches well with the tendency of bandgap in Fig. 1d and e. Meanwhile, as enlarging the hydrogenation time, the integrated current density values of hydrogenated Cs$_2$AgBiBr$_6$ PSCs raise from 1.02 mA/cm$^2$ to 7.57 mA/cm$^2$ and further to 11.29 mA/cm$^2$ (Fig. 3e). The

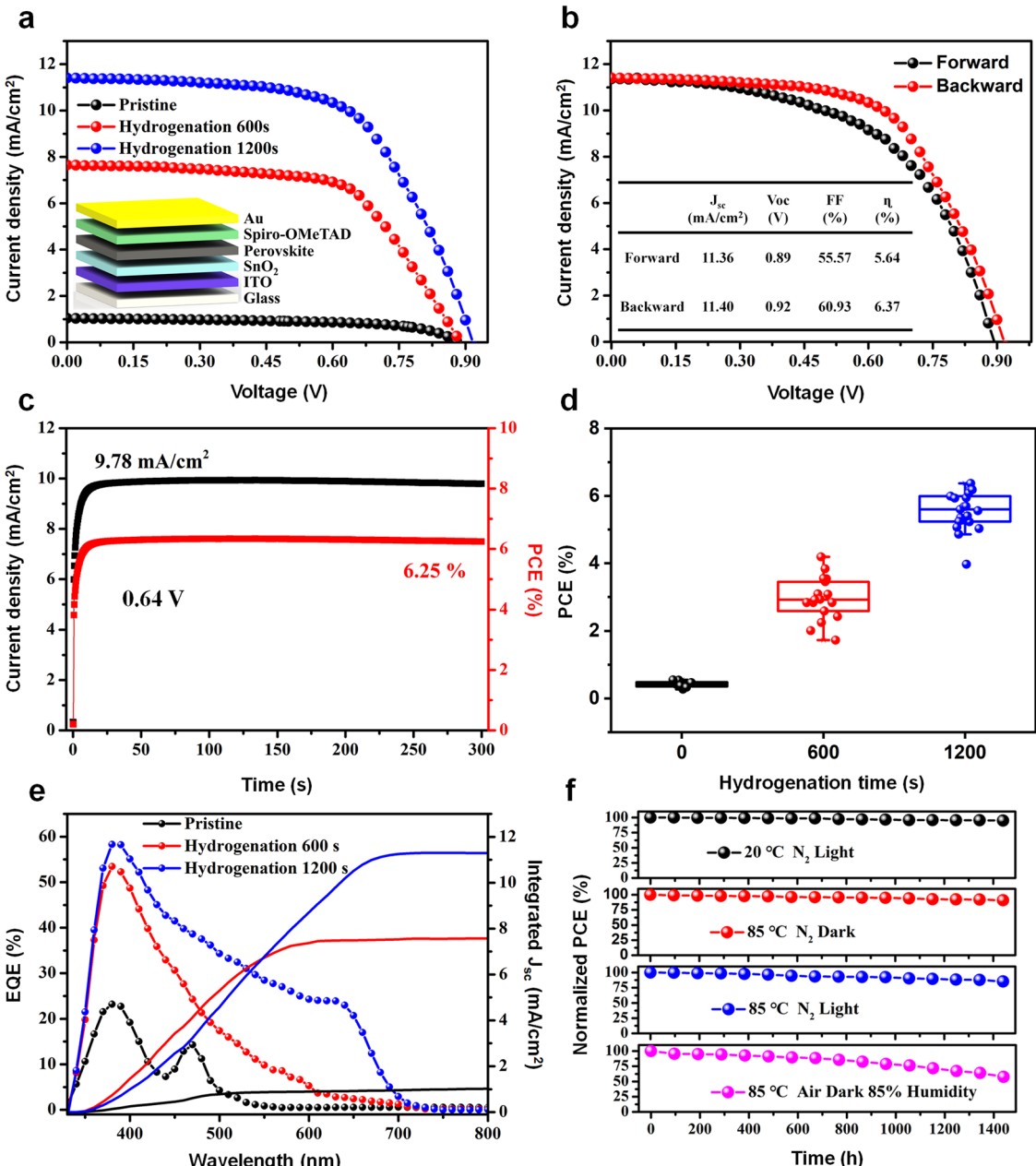

**Fig. 3 Photoelectrical performance and environmental stability of hydrogenated Cs₂AgBiBr₆ perovskite solar cell. a** Current density-voltage (*J-V*) curve of the champion Cs₂AgBiBr₆ PSCs with different hydrogenation time (0 s, 600 s and 1200 s). Inset shows the schematic of the layered perovskite solar cell. **b** *J-V* curves under reverse and forward bias of 1200 s hydrogenated Cs₂AgBiBr₆ PSC. **c** Steady-state photocurrents of 1200 s hydrogenated Cs₂AgBiBr₆ PSCs at bias voltages of 0.64 V near the maximum power output. **d** The average photoelectric conversion efficiency (PCE) distribution of Cs₂AgBiBr₆ PSCs with different hydrogenation time (0 s, 600 s and 1200 s). The outliers, middle line, upper/lower box limits and upper/lower whiskers in the box plot indicate the single points, median, 25th/75th quartiles, and maximum/minimum, respectively. **e** EQE spectrum of and integrated current density of the Cs₂AgBiBr₆ PSCs with different hydrogenation time (0 s, 600 s and 1200 s). **f** Long-term stability of 1200 s hydrogenated Cs₂AgBiBr₆ PSCs under light illumination, 85 °C, 85 °C plus light illumination and double 85 condition of 85% humidity at 85 °C, respectively.

integrated current density values with different hydrogenation time are well consistent with the corresponding $J_{sc}$ values determined by the *J-V* curves in Fig. 3a.

After hydrogenation treatment, the photoelectric properties of hydrogenated Cs₂AgBiBr₆ PSCs have been effectively improved. In order to understand the environmental stability of hydrogenated Cs₂AgBiBr₆ PSCs, the devices were treated in nitrogen at 20 °C with light illumination, and at 85 °C without or with light illumination for 1440 h, respectively (Fig. 3f). After these environmental treatments, they maintained nearly 95%, 91% and 84% of initial PCE, respectively. However, when storing in air with 85% relative humidity at 85 °C for 1440 h, the PCE of hydrogenated Cs₂AgBiBr₆ PSC devices reduced near 42% as compared with the initial one (Fig. 3f). All these results indicate that the hydrogenated Cs₂AgBiBr₆ PSCs have excellent thermo-stability and light illumination stability in nitrogen environment or when encapsulated to isolate moisture, that could promote the practical application of hydrogenated Cs₂AgBiBr₆ PSC devices.

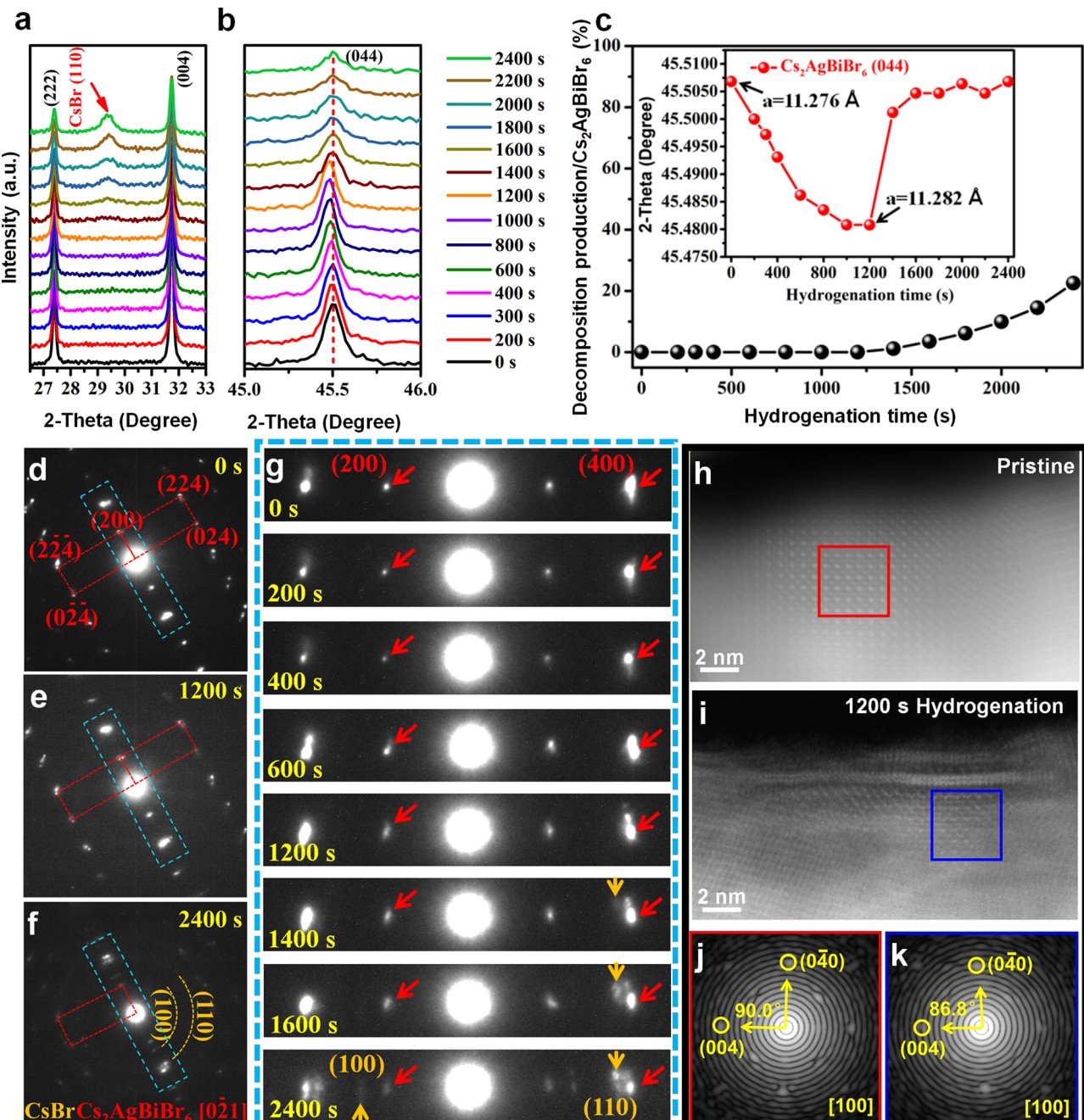

**Fig. 4 Analyses on the evolution of Cs$_2$AgBiBr$_6$ films during hydrogenation treatment. a** XRD spectra indicating the generation of CsBr (110) peak (marked by the red arrow) with different hydrogenation time. **b** XRD spectra of the Cs$_2$AgBiBr$_6$ (044) peak under different hydrogenation time. **c** Evolution tendency for the XRD area proportion of CsBr (110) peak as compared with the ones of Cs$_2$AgBiBr$_6$, inset shows the peak shift of Cs$_2$AgBiBr$_6$ (044) to the low XRD angle during the hydrogenation treatment. **d–g** Quasi in situ observation for the structural evolution of hydrogenated Cs$_2$AgBiBr$_6$ film, red and orange markers indicate the Cs$_2$AgBiBr$_6$ and CsBr phases, respectively. The detailed evolution process of the areas marked with baby blue rectangles in Fig. 4d–f were listed in Fig. 4g. **h–k** HAADF images of pristine and hydrogenated Cs$_2$AgBiBr$_6$ films, and the corresponding Fast Fourier transform (FFT) images for the areas marked by red and blue boxes.

## Discussion

In order to comprehensively understand the evolution of Cs$_2$AgBiBr$_6$ crystal structure during the hydrogenation treatment, the X-ray Diffraction (XRD) spectra of Cs$_2$AgBiBr$_6$ perovskite films with different hydrogenation time were measured as shown in Supplementary Fig. 10 and Fig. 4a. When the hydrogenation treatment time is <1200 s, the XRD peak of Cs$_2$AgBiBr$_6$-(044) was slightly offset with the increase of hydrogenation time (Fig. 4b). And a low angle shift of Cs$_2$AgBiBr$_6$-(044) XRD peak from initial

45.506°–45.480° indicates a lattice expansion of about $\Delta a = 0.006$ Å at the hydrogenation time of 1200 s (Fig. 4c). However, when increasing the hydrogenation treatment time larger than 1400 s, a new XRD peak at 29.375° appears and enhances gradually, which could be labeled as CsBr (110) in Fig. 4a and confirmed by transmission electron microscopy (TEM) in Fig. 4f and g. In addition, BiBr ($\bar{3}$12) peak may coexist and overlap with CsBr (110) peak as discussed in the caption of Supplementary Fig. 10. To further confirm this phenomenon,

quasi in situ TEM observation for the hydrogenated $Cs_2AgBiBr_6$ was shown in Fig. 4d–g (beam damage was considered as shown in Supplementary Fig. 11, detail see Methods). Initially, the main diffraction spots in the selected area electron diffraction (SAED) pattern of $Cs_2AgBiBr_6$ film were determined as along [$0\bar{2}1$] zone axis (Fig. 4d). As increasing the hydrogenation time from 0 s to 1200 s, the SAED patterns can still be identified as $Cs_2AgBiBr_6$ (Fig. 4e), but the diffraction spots gradually dim and expand (marking by the rad arrows in Fig. 4g), which indicates the lattice expansion as the ones in Fig. 4c. When enlarging the hydrogenation time up to 1400 s, CsBr (110) spot began to emerge near the $Cs_2AgBiBr_6$ ($\bar{4}00$) diffraction spot (marking by the orange arrows in Fig. 4g). Meanwhile, the morphology of the $Cs_2AgBiBr_6$ film changed obviously (Supplementary Fig. 12). And when the hydrogenation time reached 2400 s, most of the perovskite film decomposed (Fig. 4f), the XRD pattern of polycrystal CsBr and ion chromatography signal of HBr were much more clear (Supplementary Fig. 10 and Supplementary Fig. 13, Methods). It should be pointed out that, the champion PCE of hydrogenated $Cs_2AgBiBr_6$ PSC presents at the 1200 s hydrogenated samples (Fig. 3a and b). At this moment, $Cs_2AgBiBr_6$ film only processes a lattice distortion due to the atomic hydrogen insertion (Fig. 4h–k) and no impurity phase peak could be detected. So, it is reasonable to conclude that the introducing of hydrogen atom into $Cs_2AgBiBr_6$ lattice (not the decomposition intermediates) would regulate down the bandgap of perovskite film and increase the light harvest for high PCE of PSCs. In addition, to quantify the evolution tendency of $H^*$ concentration associating with the lattice expansion of $Cs_2AgBiBr_6$ film, crystal lattice parameter calculated by first-principle calculations were summarized in Supplementary Table. 4, from which we could deduce that the average doping concentration of $H^*$ should be lower than 0.3125 at.% (1/320 of atoms) in 1200 s hydrogenated $Cs_2AgBiBr_6$.

For the effect of $H^*$ on chemical state of $Cs_2AgBiBr_6$ film, X-ray photoelectron spectroscopy (XPS) was used as shown in Fig. 5a–c and Supplementary Fig. 14. For the pristine $Cs_2AgBiBr_6$ film, only one Cs $3d_{3/2}$ XPS peak at 739.2 eV was present. However, after hydrogenating the $Cs_2AgBiBr_6$ films for 600 s and 1200 s, the Cs $3d_{3/2}$ peak divided into two peaks: one shifts to a low binding energy of 738.7 eV (600 s) and then to 738.6 eV (1200 s), the other one shifts to a high binding energy from 739.2 eV (0 s) to 740.6 eV (600 s) and then to 741.1 eV (1200 s) (Fig. 5a). For Ag $3d_{3/2}$ peak, the increase of hydrogenation time would induce an increase of binding energy from 374.0 eV to 374.1 eV and further to 374.2 eV (Fig. 5b). However, the major peak of Bi $4f_{5/2}$ in $Cs_2AgBiBr_6$ films keeps at around 164.7 eV (Fig. 5c), but a small peak appears at around 162.2 eV and 162.0 eV after 600 s and 1200 s hydrogenation treatment. In addition, the binding energy position of Br 3d in $Cs_2AgBiBr_6$ films almost unchanged after 600 s and 1200 s hydrogenation treatment (Supplementary Fig. 14).

Although the hydrogenation process affects the chemical environment of Cs, Ag and Bi atoms in $Cs_2AgBiBr_6$ lattice, the occupation position of $H^*$ in the crystal lattice of $Cs_2AgBiBr_6$ is still unknown. To settle this problem, DFT calculation with HSE + SOC band structures of host $Cs_2AgBiBr_6$ is used to simulate the possible $H^*$ positions in $Cs_2AgBiBr_6$ lattice as shown in Fig. 5d and Supplementary Fig. 15. There exists three different structure units in $Cs_2AgBiBr_6$ lattice for $H^*$ incorporation, including: Ag-Br-Cs hexahedron (named as $H_1$), Bi-Br-Cs hexahedron ($H_2$), and Cs-Br-Cs octahedron ($H_3$) sites as shown in Fig. 5d and Supplementary Fig. 16. Here the $H^*$ may be a substitutional atom at the Br site in $H_n$(Br) (where $n = 1, 2,$ or 3) (Supplementary Fig. 16a–c) or as an interstitial hydrogen in $H_n$(in) polyhedrons (insets in Fig. 5d). However, for the hydrogen substitution on Br site, the bandgap for all $H_{1/2/3}$(Br)

configurations almost unchanged as compared with the host $Cs_2AgBiBr_6$ (Supplementary Fig. 15 and Supplementary Fig. 16). The inconsistency between theoretical simulation results and experimental results (Fig. 1d) indicates that the substitution of $H^*$ at the Br sites is not the main contribution for the optimization of bandgap in hydrogenated $Cs_2AgBiBr_6$.

For the interstitial doping of hydrogen (as the form of $H^*$ or $H_2$) in $Cs_2AgBiBr_6$ structure, there are various different positions, as shown in the $H_1$(in), $H_2$(in) and $H_3$(in) structures for the form of $H^*$ in Fig. 5d, Supplementary Fig. 17 and the $H_{2\text{-}1}$(in), $H_{2\text{-}2}$(in) and $H_{2\text{-}3}$(in) structures for the form of $H_2$ in Supplementary Fig. 18. However, for hydrogen molecule doping into the $Cs_2AgBiBr_6$ lattice in Supplementary Fig. 18, they are apparently not consistent with the experimental results (Fig. 1d). Therefore, we need only considering on the three different structures of $H_1$(in), $H_2$(in) and $H_3$(in) structures. For the formation energies of $H_1$(in), $H_2$(in) and $H_3$(in) structures, though $H_3$(in) has the highest value, the difference between them are small, indicating the formation of $H_1$(in), $H_2$(in) and $H_3$(in) in hydrogenated $Cs_2AgBiBr_6$ are both possible (Fig. 5d). For the host $Cs_2AgBiBr_6$, its bandgap (by using the HSE + SOC model, Supplementary Fig. 15) has been determined to be indirect with a value of 1.98 eV, which is close to the experimental one of 2.18 eV in Fig. 1). However, for the three different types of $H_n$(in) structures (Fig. 5d), interbands are formed between the upper VB and lower CB, leading to the decreasing of effective bandgaps compared with the host $Cs_2AgBiBr_6$ (Supplementary Fig. 17), which is consistent with the experimental result (Fig. 2e). In addition, we also noticed that hydrogenation treatment of the $Cs_2AgBiBr_6$ film induced almost no defect state increasing between its valence and conductive bands (Supplementary Fig. 19). And the physical model for the energy level positions of $H_n$(in) structures are derived as shown in Fig. 5e. On one hand, H-1s orbital couples with cation and forms the bonding state (Supplementary Fig. 17), which is higher than VBM (Fig. 5e); on the other hand, H-1s orbital couples with anion and forms antibonding state (Supplementary Fig. 17), which is higher than CBM (Fig. 5e). This physical model is well matched with the experimental energy-level diagram in hydrogenated $Cs_2AgBiBr_6$ (Fig. 2e, f).

In addition, the variations of charges on Ag, Cs and Bi atom sites in different $H_n$(in) structures are investigated by bader analysis and compared with the one in host structure, as shown in Fig. 5f. The decreased bader charge of Ag atom in $H_1$(in) means Ag losing some electrons to the surrounding atoms and leading to the increasing of Ag 3d binding energy. This is in agreement with the variation tendency of Ag 3d peak position shift in XPS results (increasing of binding energy in Fig. 5b and g), which indicates the existence of $H_1$(in) in hydrogenated $Cs_2AgBiBr_6$. However, the increasing bader charge of Bi atom in $H_2$(in) suggests an electron gain of Bi atom from the surrounding atoms (Fig. 5f), resulting in the decreasing binding energy of Bi 4f. This is consistent with the decrease of Bi 4f binding energy in XPS results (Fig. 5c and g) and reveals the formation of $H_2$(in) in hydrogenated $Cs_2AgBiBr_6$. What's more, the existence of $H_1$(in) and $H_2$(in) structures would induce the decrease of bader charges (Fig. 5f) and increase of binding energy of Cs atoms, which is well matched with the peak position increasement of Cs 3d in XPS results (Fig. 5a and g). On the other hand, the presence of $H_3$(in) induces an increase of bader charge of Cs atoms, which corresponds to a decreasing binding energy of Cs 3d (Fig. 5a and g). So, by analyzing the changes of bader charges and comparing them with XPS results (Fig. 5), we confirm that the interstitial doping of $H^*$ in $H_1$(in), $H_2$(in) and $H_3$(in) structures do exist in the hydrogenated $Cs_2AgBiBr_6$.

In this paper, we have developed a hydrogenation method to modify the material properties of $Cs_2AgBiBr_6$ film. After

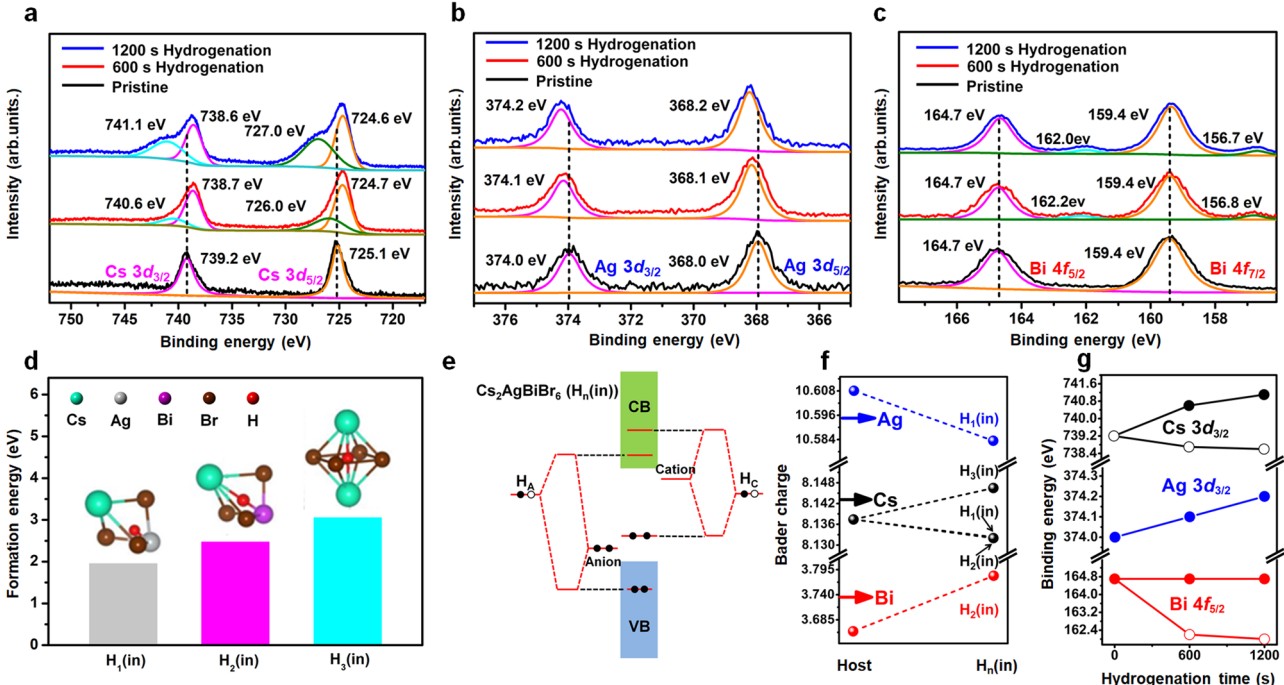

**Fig. 5 Analyses on the chemical environment of Cs$_2$AgBiBr$_6$ during hydrogenation treatment. a–c** X-ray photoelectron spectroscopy (XPS) spectra of Cs 3$d$, Ag 3$d$ and Bi 4$f$ in Cs$_2$AgBiBr$_6$ films with different hydrogenation time (0 s, 600 s and 1200 s). **d** The formation energies of H$_1$(in), H$_2$(in) and H$_3$(in). Here, H$_n$(in), where $n = 1$, 2, or 3, presents the H atom in the interstitial position of the H$_n$ polyhedrons surrounded by Cs-Br-Ag (H$_1$), Cs-Br-Bi (H$_2$), and Cs-Br-Cs (H$_3$), respectively. Cs, Ag, Bi, Br and H atoms are represented by cyan, light gray, purple, brown and red dots, respectively. **e** Schematic band coupling models of H$_n$(in) showing energy level positions when the H 1$s$ orbital is coupled to anion (forming donor) and cation (forming acceptor) levels (H$_A$ (coupled from interstitial H$_n$(in) and anion) and H$_C$ (coupled from interstitial H$_n$(in) and cation)). **f** Bader charge variations of Ag, Bi, and Cs atoms in the host and next to the interstitial H$^*$ in the H$_n$(in) polyhedrons. **g** The variation tendency of binding energy of Cs 3$d_{3/2}$, Ag 3$d_{3/2}$ and Bi 4$f_{5/2}$ peaks as the increase of hydrogenation time.

hydrogenating treatment of the Cs$_2$AgBiBr$_6$ film from 0 s to 1200 s, the light absorption at the wavelength above 496 nm increases apparently, which corresponds to a decrease of bandgap from 2.18 eV to 1.64 eV. At the same time, the carrier concentration of the hydrogenated Cs$_2$AgBiBr$_6$ films increases from $1.42 \times 10^{12}$ cm$^{-3}$ to $5.96 \times 10^{12}$ cm$^{-3}$, and the carrier mobility and lifetime have also been largely improved from 1.71 cm$^2$V$^{-1}$s$^{-1}$ and 18.85 ns, to 9.28 cm$^2$V$^{-1}$s$^{-1}$ and 41.86 ns, respectively. Based on this, the hydrogenated Cs$_2$AgBiBr$_6$ perovskite films are fabricated into solar cells, and the highest PCE of 6.37% is achieved, which is a record high efficiency of Cs$_2$AgBiBr$_6$-based PSC. First-principle calculations confirm that hydrogen atoms are doped into the interstitial sites of Cs$_2$AgBiBr$_6$ lattice, and the band couplings between hydrogen atoms and anion/cation alter the valence/CB levels, which agrees well with the experimental results. All these findings provide an effective lattice engineering strategy for preparing high-efficient and bandgap tunable lead-free inorganic PSCs, which are not only environment-friendly for optoelectronic device applications, but also remarkable stable in appropriate working or storage environments. And this discovery may offer a good opportunity for exploring the next generation PSC and other optoelectronic devices.

## Methods

**Materials.** The experimental materials such as tin oxide (SnO$_2$, 15%), cesium bromide (CsBr, 99.9%), silver bromide (AgBr, 99.5%) and bismuth bromide (BiBr$_3$, 99%) were purchased from Alfa Aesar. Dimethylsulfoxide (DMSO, anhydrous 99.9% Sigma-Aldrich), isopropanol (IPA 99.5%, Aladdin), acetonitrile (99.8%, Sigma-Aldrich), 4-tertbutylpyridine (96%, Sigma-Aldrich), bis (trifluoromethane) sulfonimide lithium salt (99%, Xi'an Polymer Light Technology Corp), chlorobenzene (99%, Aladdin) and spiro-OMeTAD (99.5%, Xi'an Polymer Light Technology Corp) were used without further purification.

**Preparation of perovskite precursor solutions.** In this work, we adopt a solution spin-coating method[4] to prepare high quality Cs$_2$AgBiBr$_6$ films. As we all known, the solubility of Cs$_2$AgBiBr$_6$ in dimethylsulfoxide (DMSO) solution is about 0.6 mol/L, here we choose DMSO as the precursor solvent[20]. The Cs$_2$AgBiBr$_6$ perovskite precursor solution (0.6 mol/L) was prepared by dissolving 212.81 mg CsBr, 187.77 mg AgBr and 448.68 mg BiBr$_3$ into 1 mL of anhydrous solvent of DMSO. In order to ensure sufficient solubility and prepare high quality Cs$_2$AgBiBr$_6$ perovskite film, the perovskite precursor solution was stirred at 70 ℃ in glove box for 12 h until the solutes were completely dissolved into the DMSO solvent and filtered through 0.45 μm poly tetra fluoro ethylene (PTFE) filtering membrane.

**Determination of the diffusion coefficient of atomic hydrogen in Cs$_2$AgBiBr$_6$ film.** As shown in Eq. 2, the evolution of $\frac{C}{C_s}$ values could be deduced as a change of hydrogenation time and atomic hydrogen diffusion coefficient (Supplementary Fig. 4). However, due to the thickness of Cs$_2$AgBiBr$_6$ film is about 140 nm (Supplementary Fig. 4f), so the determination of $\frac{C}{C_s}$ values at the top surface and bottom of a 140 nm thick Cs$_2$AgBiBr$_6$ film is critical important as shown in Supplementary Table 3. However, in practice, it is very difficult to quantify the atomic hydrogen concentration at the top surface and bottom of the hydrogenated Cs$_2$AgBiBr$_6$ film, which inversely limits the analysis on the accurate diffusion coefficient of atomic hydrogen in Cs$_2$AgBiBr$_6$ film. To settle this problem, we firstly extract the $\frac{C_f}{C_{s'}}$ values from the experimental optical images in Fig. 2a and compare them with the simulated values $\frac{C}{C_s}$ in Supplementary Table 3. As shown above, the optical absorption properties of hydrogenated Cs$_2$AgBiBr$_6$ film increases as the increasing of hydrogenation time (actually the hydrogen concentration increase), and film became much darker (Fig. 2a). From this point of view, it is assumed that the blackness values of the front and back side optical images (in Fig. 2a) are directly proportional to the average hydrogen concentration near the top and bottom surface of hydrogenated Cs$_2$AgBiBr$_6$ film, respectively (as shown by $C_s'$ and $C'$ in Supplementary Fig. 4g). However, the hydrogen concentration is gradient distributed from the top surface to the bottom of film (Fig. 2b), this means that the experimental $\frac{C_f}{C_{s'}}$ values must be larger than the simulated $\frac{C}{C_s}$ values ($\frac{C_f}{C_{s'}} > \frac{C}{C_s}$) (Supplementary Fig. 4g).

According to the experimental $\frac{C_f}{C_{s'}}$ value of 0.205 at the hydrogenation time of 600 s (Supplementary Table 3), the simulated $\frac{C}{C_s}$ value should be less than 0.205 and we can find that only a diffusion coefficient smaller than $1 \times 10^{-13}$ cm$^2$/s

$(\frac{C}{C_s} = 0.203)$ is appropriate. Meanwhile, such a diffusion efficient of $1 \times 10^{-13}$ cm$^2$/s is applicable to the $\frac{C_{l'}}{C_{s'}}$ and $\frac{C}{C_s}$ values at the hydrogenation time of 1200 s $(\frac{C_{l'}}{C_{s'}} = 0.405 > \frac{C}{C_s} = 0.366)$, 1800 s $(\frac{C_{l'}}{C_{s'}} = 0.834 > \frac{C}{C_s} = 0.460)$, 2400 s $(\frac{C_{l'}}{C_{s'}} = 0.963 > \frac{C}{C_s} = 0.522)$ and 3000 s $(\frac{C_{l'}}{C_{s'}} \approx 1.000 > \frac{C}{C_s} = 0.571)$. It should be pointed out that, as the hydrogenation time increasing up to 3000 s, the front and back sides color of the hydrogenated Cs$_2$AgBiBr$_6$ film are both black, which takes a $\frac{C_{l'}}{C_{s'}} \approx 1.000$. It indicates that a large number of hydrogen atoms must have reached to the bottom of the Cs$_2$AgBiBr$_6$ film after 3000 s hydrogenation treatment. However, from the data in Supplementary Table 3, if the diffusion efficient is $<1 \times 10^{-14}$ cm$^2$/s, then the $\frac{C}{C_s}$ value equals to 0.074 even after a 3000 s of hydrogen diffusion. From all the analyses above, we can conclude that the diffusion coefficient $D$ of atomic hydrogen in Cs$_2$AgBiBr$_6$ film must be located in the range between $1 \times 10^{-14}$ cm$^2$/s and $1 \times 10^{-13}$ cm$^2$/s.

**Device fabrication**. Inorganic lead-free double PSCs with typical planar structure ITO/SnO$_2$/perovskite/spiro-OMeTAD/Au were fabricated as follows: The ITO glass substrates were washed by ultrasonic cleaning in acetone, ethanol, detergent and deionized water for 30 min, respectively. After drying, the ITO glass substrates were treated by an oxygen plasma for 10 min to improve the hydrophilicity of the substrate surface. The original solution of SnO$_2$ was mixed with deionized water in the volume ratio of 1:5. SnO$_2$ solution of 50 μL was dynamically spin-coated on the clean ITO glass substrates at 3000 rpm for 30 s. Then the SnO$_2$ ETL film was annealed at 150 °C for 30 min in air. In order to improve the surface coverage and film quality, a lot of efforts were made in our research such as: increasing the spin-coating temperature[53] to 40 °C and using isopropanol (IPA) solution as anti-solvent during the spin-coating process as shown in Fig. 1a. Cs$_2$AgBiBr$_6$ perovskite films were deposited in the glove box full of N$_2$ using traditional solvent-quenching method with IPA as the antisolvent. 50 μL Cs$_2$AgBiBr$_6$ precursor solution was dynamically spin-coated at 40 °C on the cleaned ITO glass substrates at 5000 rpm for 50 s, and 300 μL antisolvent (IPA solvent) was spin-coated at 10 s, which was just before the atomization point. Then the film was annealed at 290 °C for 5 min. Then, the perovskite films were plasma treated for different times.

Spiro-OMeTAD was used as the HTL. To prepare the Spiro-OMeTAD solution, 45 μL Li-TFSI solution (170 mg/mL acetonitrile), 10 μL 4-TBP and 90 mg Spiro-OMeTAD were mixed into 1 mL chlorobenzene solvent and then stirred in a dark environment in N$_2$ glove box until fully dissolved. After that, 50 μL Spiro-OMeTAD solution was dynamically spin-coated on the surface of the perovskite film at 3000 rpm for 30 s in N$_2$ glove box. After the spin-coating, it was placed in oxygen atmosphere and kept away from light for 24 h to make Spiro-OMeTAD layer fully oxidized. Au electrode was coated on the Spiro-OMeTAD HTL by thermal evaporation. The interface between the electrode film and the HTL was well combined. The electrode film was uniform with the thickness of about 60 nm.

**Plasma treatment on the perovskite film**. In order to doping the hydrogen atoms into Cs$_2$AgBiBr$_6$ film, the film was hydrogenated through a plasma treatment (Soft Plasma Cleaner SPC 150) with the source power of 30 W in Ar and H$_2$ mix (80 vt.% H$_2$) gas. For the plasma equipment, with a molecular pump keeps the stable gas pressure in the working condition, the plasma power is of lower than 40 Watts and equipped with radio frequency (RF) ion source of 13.56 MHz. Indicating that the sample could be considered as a uniformly exposure of steady plasma environment[54].

Normally, one of the most important feature of "cold" plasma is that the electron temperature is lower than 10 eV[55]. According to the average electron temperature values listed in Supplementary Table. 5, the plasma treatment in our experiment belongs to low temperature plasma. To intuitively observe the local temperature during the working condition in plasma chamber, the irreversible surface temperature indicating strips (ISTIS, Thermax, temperature error of ±1 °C) were used to determine the temperature as shown in Supplementary Fig. 20a–c, from which the working temperature was finally determined to be lower than 37 °C. In addition to the environment temperature of the thermal steady state of the plasma chamber, the temperature of the chemical events at the atomic scale should also be taken into account. To describe the chemical interaction between plasma gas and Cs$_2$AgBiBr$_6$ film, the maximum energy transfer $E_T$ from highly energetic hydrogen radicals to the surface atom of Cs$_2$AgBiBr$_6$ should be considered [Eq. 3],

$$E_T = \frac{2ME(E + 2mc^2)}{(m + M)^2 c^2 + 2ME} \quad (3)$$

here, $E$ is kinetic energy of the incoming plasma excited hydrogen radical; $m$, $M$ is the mass of hydrogen atom and the chemical elements in Cs$_2$AgBiBr$_6$; $c$ is the velocity of light ($3.00 \times 10^8$ m/s). If the kinetic energy $E$ is much smaller than $mc^2$ ($mc^2 = 9.40 \times 10^8$ eV for the hydrogen atom) [Eq. 4], the equation could be simplified as[56]

$$E_T = E \frac{4Mm}{(m + M)^2} \quad (4)$$

In this work, the maximum kinetic energy of radicals in hydrogen plasma is 9.3 eV (Supplementary Table 5), which is much smaller than $mc^2$. So, the energy transfer from the hydrogen radicals to the surface atoms in Cs$_2$AgBiBr$_6$ could be deduced from Eq. 4. Based on this, the maximum energy transfer $E_F$ to the surface atoms of Cs, Ag, Bi and Br could be determined as 0.28 eV, 0.34 eV, 0.18 eV and 0.46 eV, respectively, which are obviously lower than the bond dissociation energies of Cs-Br (4.03 eV), Bi-Br (2.49 eV) and Ag-Br (2.91 eV) in Cs$_2$AgBiBr$_6$ (Supplementary Table 6)[57]. Therefore, the calculation results indicate that the low temperature hydrogen plasma treatment is hard to break the chemical bonds in Cs$_2$AgBiBr$_6$.

However, the Cs$_2$AgBiBr$_6$ lattice is stable only when the hydrogenation treatment is <1200 s. The decomposition of hydrogenated Cs$_2$AgBiBr$_6$ phase occurs after 1400 s hydrogenation treatment (Fig. 4), which is attributed to the intrinsic structural instability of hydrogenated Cs$_2$AgBiBr$_6$. As increasing the concentration of atomic hydrogen into Cs$_2$AgBiBr$_6$ lattice, the formation energy of perovskite does process a raising (Fig. 5d).

**Determination of perovskite film decomposition products by ion chromatography**. Bromine-anion calibration standard solutions were prepared by diluting the HBr stock solution with deionized water to the desired concentration. The eluent was prepared by adding 20 mL 0.24 mol/L Na$_2$CO$_3$ water solution and 10 mL 0.3 mol/L NaHCO$_3$ water solution into 1000 mL ultrapure water. All solutions were filtered using a 0.45 μm filter before use. A commercial Metrohm 883 Basic IC plus instrument equipped with anion-separation column (Metrosep A Supp 5) was used for obtaining the ion chromatogram.

**Characterization of solar cells and films**. We measured $J$-$V$ curves of PSC devices in air with Keithley 2400 Source under AM 1.5 (100 mW cm$^{-2}$, xenon lamp, Newport) irradiance level. All the $I$-$V$ measurements were carried out in ambient air environment. The area of each device is 0.2 cm$^2$. During measuring, a 0.04 cm$^2$ non-reflective mask was used to define the accurate active cell area with the testing range of reverse scanning voltage from 1.2 V to −0.2 V (step 0.02 V). The drive-level capacitance profiling measurement was performed at ac frequency of 10 kHz (total carrier density) by subtracting frequency and 500 kKz (free carrier density) by using Semiconductor Device Analyzer (Keysight B1500A). Steady-state and transient-state PL spectra were obtained by using FLS-1000 fluorescence spectroscopy (Edinburgh Instruments) at 430 and 470 nm excitation wavelength, respectively. The external quantum efficiencies and integrated current of PSCs were measured by using Zolix SCS10-X150-DSSC. The optical absorption spectrum was measured in air by an ultraviolet-visible near-infrared spectrophotometer (UV-Vis) (Hitachi, UH-4150). The XRD spectra of the perovskite films were obtained with Bruker D8 Advance X-ray diffractometer using Cu K$_\alpha$ radiation ($\lambda = 1.5418$ Å). Scanning electron microscope (Helios Nanolab 600i) operated at 5 kV was used to characterize the microstructure of films and cross-section of PSC. The carrier lifetime was obtained by using B411 (FLS-1000). The XPS and UPS spectrums were measured through XPS system (ESCALAB 250Xi). The electrical properties were measured with four-point probe method by using Hall-effect measurement system (Eastchanging ET 9000).

**The preparation and characterization of transmission electron microscopy (TEM) samples**. The TEM sample of Cs$_2$AgBiBr$_6$ perovskite devices were prepared through Focused Ion Beam (FIB) system (Helios Nanolab 600i) operating at 2–30 kV. In order to make sure the cleanliness of the device surface, the sample was first cleaned by N$_2$ gas gun before putting into the FIB system. Firstly, we selected one flat area and deposited 1.2 μm thick Pt layer (200 nm e-beam deposition with 5 kV, 2.7 nA followed by 1 μm ion beam deposition with 30 kV, 30 pA) onto the surface of device, which could protect the sample from the harm of Ga ion beam damage. Then, the specimen was crosscut and thinned to about 200 nm from 1.5 μm by using 30 kV Ga ion beam. Lastly, in order to remove the surface damage layer, a final polishing was performed first using 5 kV, 15 pA and further using 2 kV, 23 pA Ga ion beam. TEM sample of Cs$_2$AgBiBr$_6$ films was prepared by transfer of as-grown Cs$_2$AgBiBr$_6$ films onto a TEM grid. Samples were investigated by using probe spherical aberration corrected transmission electron microscope (FEI Titan G2) with 300 kV accelerating voltage and gain the images using scanning transmission electron microscopy-high angle annular dark field (STEM-HAADF) detectors. The microscopic chemical compositions of the samples were analyzed by using high efficient "Super X" Energy Disperse Spectroscopy (EDS) detector. The energy resolution of the super-EDX is 137 eV. The EDS data were collected and processed by using Esprit 1.9 software. In order to explore the effect of electron beam on lead-free inorganic double perovskite Cs$_2$AgBiBr$_6$, in situ analysis on structure evolution under continuous electron beam was performed through SAED as shown in Supplementary Fig. 11. Based on the results in Supplementary Fig. 11, to avoid the beam damage on the TEM characterization of Cs$_2$AgBiBr$_6$ samples, the total electron dose should be controlled less than 300 e$^-$/Å$^2$. The electron dose rate was controlled to be 1 e$^-$/Å$^2$s for TEM model and the acquisition time for each TEM image is 1 s (Fig. 4d–g, Supplementary Fig. 11 and Supplementary Fig. 12). While, for the HAADF images in Fig. 4h and i, the electron dose rate is about $2 \times 10^6$ e$^-$/Å$^2$s, and the acquisition time of each pixel is about 2 μs.

**DFT calculation**. First-principles calculations were performed using DFT as implemented in the VASP code[58,59]. The electron and core interactions were

included using the frozen-core projected augmented wave approach[60]. The generalized gradient approximation of Perdew–Burke–Ernzerhof (PBE)[61] was used for the exchange correlation functional. The kinetic energy cutoff for plane-wave basis functions is set to 400 eV. K-point meshes with grid spacings of $2\pi \times 0.025$ Å$^{-1}$ or smaller were used for Brillouin zone integration. The structures of $Cs_2AgBiBr_6$ with the $Fm\overline{3}m$ space group were relaxed until the total energies converged to $10^{-4}$ eV. All the atoms are fully relaxed for the substitutional hydrogen ($H_{1/2/3}(Br)$) and interstitial hydrogens ($H_{1/2/3}(in)$) in $Cs_2AgBiBr_6$ lattice. The formation energies and Bader charges of $H_{1/2/3}(in)$ are conducted using PBE functional. The formation energy are defined as: $\Delta E_f = E(H_i) - E(host) - 0.5 \cdot E(H_2)$, where $E(H_i)$, $E(host)$ and $E(H_2)$ are the total energies of $H_i$, pure $Cs_2AgBiBr_6$ and hydrogen molecule, respectively. Whereas, a hybrid Heyd–Scuseria–Ernzerhot (HSE)[62] with a standard 25% Hartree Fock exchange was used for evaluating the bandgaps. Due to the heavy element Bi in $Cs_2AgBiBr_6$, spin-orbit coupling (SOC) effect was also taken into consideration in the calculations.

**Ageing process of perovskite films**. $Cs_2AgBiBr_6$ perovskite films could change from yellow to black after hydrogenation process, indicating the large increase in light absorption. Then we ageing the hydrogenated $Cs_2AgBiBr_6$ films at different condition including: 20 °C/$N_2$/dark, 20 °C/$N_2$/1-sun-light-illumination and 85 °C/$N_2$/dark environments. Then we take optical images of the hydrogenated $Cs_2Ag$-$BiBr_6$ films at different ageing time and gain the bandgap values of the films through UV-Vis.

**Reporting summary**. Further information on research design is available in the Nature Research Reporting Summary linked to this article.

## Data availability

The source data generated in this study are provided in the Source Data file. Source data are provided with this paper.

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

## Acknowledgements

This work was supported by the Beijing Innovation Team Building Program, China (IDHT20190503), the National Natural Science Foundation of China (11704015, 51621003, 12074016), the General Program of Science and Technology Development Project of Beijing Municipal Education Commission (KM202110005003), Beijing Natural Science Foundation (Z210016) and the National Key Research and Development Program of China (2016YFB0700700).

## Author contributions

Y.L. and M.L.S. conceived and designed the experiments. Z.Y.Z. prepared the samples, performed the measurement of X-ray diffraction, scanning electron microscopy, ultraviolet-visible absorption, hall effect, time-resolved photoluminescence, ultraviolet photoelectron spectroscopy, X-ray photoelectron spectroscopy, transmission electron microscope, current density-voltage curve and calculation the diffusion coefficient. Q.D.S., S.H.W., and F.L. did the theoretical calculations. X.L.M. prepared the perovskite film. Z.Y.Z., Y.L., M.L.S., and S.H.W. analyzed the data and co-wrote the paper. All authors discussed the results and revised the paper.

## Competing interests

The authors declare no competing interests.
