## [Peer Review File · Nature Communications]

Hydrogenated Cs₂AgBiBr₆ for Significantly Improved Efficiency of Lead-Free Inorganic Double Perovskite Solar CellREVIEWER COMMENTS

Reviewer #1 (Remarks to the Author):

In this manuscript entitled Hydrogenated Cs₂AgBiBr₆ for High-Efficient Lead-Free Inorganic Double Perovskite Solar Cell, Sui et al. present a lead-free Cs₂AgBiBr₆ double perovskite for PV applications. By controlled plasma hydrogenation, they lowered bandgap from 2.14 eV to 1.61 eV, making it suitable for solar cells. As such, they managed to improve solar-cell-efficiency to as high as 6.27%. The manuscript shows some common measurements with the first principle calculation analysis. Considering that it is the first to report over 5% efficiency for the double perovskite solar cells, I would recommend it for NC upon the following are properly addressed:

1. The manuscript claims "very stable in different working or storage environments". This claim is too general for a scientific paper. It should be specific. Particularly when it also says "the hydrogenated Cs₂AgBiBr₆ films exhibit extraordinary stability under light illumination or at temperature of 85 oC in nitrogen environment". The authors should provide stability of solar cells, not just the "Cs₂AgBiBr₆ films". The stability of the individual film means too little for a complicated solar cell device.
2. I suggest the authors provide stability of the solar cell (1) under light illumination, (2) 85% relative humidity at 85 oC.
3. The authors use the traditional solar cell structure ITO/SnO₂/perovskite/spiro-OMeTAD/Au. This structure is designed for the Pb-based solar cells. Does the energy levels of the double perovskite also match well with SnO₂ and spiro-OMeTAD? Please provide measurements.
4. How does EQE match with the optical bandgap measurement?
5. The language of the manuscript is very poor. Please have it edited before the next submission.
6. Can authors provide a larger area cell with J-V measurement?

Reviewer #2 (Remarks to the Author):

In this paper, the authors report on bandgap reduction of Cs₂AgBiBr₆ double perovskite through hydrogen plasma treatment. With an appropriate duration of treatment, the bandgap was reduced to 1.61 eV from 2.14 eV. Through DFT calculations, the authors identified interstitial H as the possible cause for the observed bandgap reduction. The incorporation of interstitial hydrogen not only reduce the bandgap, but also improve carrier mobility and lifetime. As a result, the power conversion efficiency (PCE) of Cs₂AgBiBr₆ perovskite solar cells improved significantly, with the highest PCE of 6.27%. These results seem interesting. I have the following comments:

1. The PCE of 6.27% is commendable for Cs₂AgBiBr₆ PSCs, but it is not appropriate to call it "High-efficient." Perhaps the authors may use some terms like "significantly improved efficiency."
2. The authors should report the operational stability of the Cs₂AgBiBr₆ PSCs under one sun illumination, rather than just the films.
3. The solar cells should be better measured and reported. For example, EQE curve should be provided. J-V curves measured under reverse and forward bias should be reported. The J-V curves for the hydrogenated PSCs show a V shape, suggesting all fill factors and, therefore, efficiencies could be overestimated. Authors are suggested to measure and report power outputs, i.e., measure photocurrents at photovoltages near the maximum power output.
4. The determination of bandgap from absorption spectra is not convincing. There are significant below bandgap absorptions with almost the same onset for all Cs₂AgBiBr₆ including the pristine one. What is the origin of the below bandgap absorption? Is it due defects such as local disorder/lattice distortion? It seems H incorporation is just to increase the density of such defects.
5. Can the disorder or lattice distortion induced by H incorporation contribute to the reduced band gap?
6. What is the estimated H concentration in the hydrogenated Cs₂AgBiBr₆? Can H₂ molecules form in Cs₂AgBiBr₆?
7. For TEM study, particularly for HAADF, did electron beam damage the sample?

Reviewer #3 (Remarks to the Author):

The manuscript entitled "Hydrogenated Cs₂AgBiBr₆ for High efficient Lead free Inorganic Double Perovskite solar cell" reports a new class of materials to be used as solar cells after being treated in hydrogen plasma. The plasma treatment durations is a parameter in the study. These devices are manufactured by spin coating from a DMSO solution of the precursor compounds followed by steps of annealing and hydrogen plasma treatments. The working hypothesis of this article is that the presence of hydrogen, introduced in the structure during a plasma treatment, influences the bandgap and solar to electrical conversion efficiency of the devices. Having hydrogen species with unidentified oxidation state, i.e. a radical, an ion or an atom, DFT calculations are reported.

In order to ensure that these effects can be attributed to the "hydrogen species" in the lattice, the authors must provide details about the following:

1. XRD patterns given in the supplementary information section of the manuscript exhibit distinct formation of a new structure, identified by the authors as due to CsBr(110). Despite the authors claims that this phase forms only after 1800 s exposure, one would suspect that these species are present, but the grain size of CsBr crystallites was not large enough to be detectable by XRD at lower plasma exposure times. While the intensity of the peak due to CsBr is increasing, the intensity of the peaks that are due to the pristine perovskite crystal systematically decrease with increasing hydrogen plasma exposure period. As a result, it is strongly recommended that the authors provide the XRD patterns of the samples that were exposed to plasma for shorter periods with a thorough analysis of the peak intensities, and line widths. As far as the decomposition process is concerned starting from a Cs₂AgBiBr₆, after the formation of CsBr, the residual should have a stoichiometry of CsAgBiBr₃. As such this is a completely different material. If CsBr is detectable in the XRD pattern after a certain hydrogen exposure period, a thorough and quantitative analysis of XRD patterns of the prepared compounds is needed. Only after the assignments of the existing structures and the impurity levels are quantified, a discussion about the presence and the role of hydrogen is meaningful. The authors are strongly encouraged to reveal the role of the new chemical environments and their structure in the light absorption properties and carrier mobilities. While it is not easy to identify new structures experimentally, DFT can provide assistance of the new compounds and their XRD patterns. The XPS data provided in Figure 4 for Cs should also be reinterpreted within the context of the CsBr compound and the CsAgBiBr₃ that should form as a result.

2. The plasma treatment details are not given to see the local temperatures. If the temperatures are sufficiently high, hydrogen diffusion is going to be very rapid. Without a precise temperature distribution, it is not meaningful to extract a hydrogen diffusion coefficient based on the color profile of the films. Of course this argument is only meaningful if we have a material with intact crystal structure, a point that needs to be justified in the light of the comments made in item 1 above.

3. The concentration profile given in equation 2 is the result of the solution of Fick's second law in cartesian coordinates in one dimension. The process described in the manuscript needs to be justified whether the cartesian representation of the diffusion problem is justifiable, i.e. sample is uniformly exposed to plasma by a steady (i.e. not transient) exposure. Only after this justification, the methodology described can be used to extract hydrogen diffusion coefficient.

Once these points are clarified, the work is publishable.

Reviewer #1 (Remarks to the Author):

Reviewer #1: In this manuscript entitled Hydrogenated Cs₂AgBiBr₆ for High-Efficient Lead-Free Inorganic Double Perovskite Solar Cell, Sui et al. present a lead-free Cs₂AgBiBr₆ double perovskite for PV applications. By controlled plasma hydrogenation, they lowered bandgap from 2.14 eV to 1.61 eV, making it suitable for solar cells. As such, they managed to improve solar-cell-efficiency to as high as 6.27%. The manuscript shows some common measurements with the first principle calculation analysis. Considering that it is the first to report over 5% efficiency for the double perovskite solar cells, I would recommend it for NC upon the following are properly addressed:

Reply: We are grateful to receive reviewer's valuable comments, all the suggestions have been responded one-by-one as follows.

1. The manuscript claims "very stable in different working or storage environments". This claim is too general for a scientific paper. It should be specific. Particularly when it also says "the hydrogenated Cs₂AgBiBr₆ films exhibit extraordinary stability under light illumination or at temperature of 85 °C in nitrogen environment". The authors should provide stability of solar cells, not just the "Cs₂AgBiBr₆ films". The stability of the individual film means too little for a complicated solar cell device.

Reply: Thanks for reviewer's valuable comments. We rephrase the sentences in the revised version to make them more scientific and accurate. In order to give a comprehensive understanding on the stability of perovskite solar cells, the fabricated devices were treated under light illumination in N₂, heating at 85 °C in N₂ without and with light illumination, and 85% relative humidity at 85 °C as shown in Fig. 3f.

2. I suggest the authors provide stability of the solar cell (1) under light illumination, (2) 85% relative humidity at 85 °C.

Reply: As reviewer's suggestions, the stability of the solar cells under different conditions is shown in Fig. 3f. After aging the 1,200 s hydrogenated Cs₂AgBiBr₆ PSC devices under light illumination, at 85 °C in N₂ without and with light illumination conditions for 1,440 hours, the PCE maintained nearly 95%, 91% and 84% of initial ones, respectively. However, when

storing in 85% relative humidity at 85 °C for 1,440 hours, the PCE of hydrogenated Cs₂AgBiBr₆ PSC devices significantly reduced to 58% of initial efficiency (Fig. 3f). All these results indicate that the hydrogenated Cs₂AgBiBr₆ PSCs present excellent thermostability and light stability, but should be packed to isolate the moisture, especially during its practical application. All discussions have been added on page 11 in revised manuscript.

3. The authors use the traditional solar cell structure ITO/SnO₂/perovskite/spiro-OMeTAD/Au. This structure is designed for the Pb-based solar cells. Does the energy levels of the double perovskite also match well with SnO₂ and spiro-OMeTAD? Please provide measurements.

Reply: Thanks for the good suggestion. In order to verify the compatibility of SnO₂ and spiro-OMeTAD as the ETL and HTL in Cs₂AgBiBr₆ PSC device, the energy level of these two layers were measured through UV-vis and UPS spectra as shown in Supplementary Fig. 6. From the results in Supplementary Fig. 6, the energy level of E_{VBM} of SnO₂ and E_{CBM} of Spiro-OMeTAD films could be determined as -4.51 eV and -2.33 eV, respectively. As compared with the energy level of hydrogenated Cs₂AgBiBr₆ film (Fig. 2f), the energy level of double perovskite matches well with SnO₂ and spiro-OMeTAD. All these results have been discussed on page 9 in the manuscript and page 6 in revised supplementary information.

4. How does EQE match with the optical bandgap measurement?

Reply: Thanks for the kind reminding. As reviewer's suggestion, external quantum efficiency (EQE) is also a good method to judge the optical bandgap of semiconductors. In this revision, EQE of Cs₂AgBiBr₆ perovskite solar cells with different hydrogenation time (0 s, 600 s and 1,200 s) are shown in Fig. 3e. Here, it could be found that the photo-response range expands gradually from 539 nm for pristine to 720 nm for 1,200 s hydrogenated Cs₂AgBiBr₆ PSC, which matches well with the results in Figs. 1d and 1e. All these results have been discussed on pages 10 and 11 in revised manuscript.

5. The language of the manuscript is very poor. Please have it edited before the next submission.

Reply: We appreciate reviewer's suggestions. We have gone through the manuscript carefully

and improved the language of the manuscript.

6. Can authors provide a larger area cell with J-V measurement?

Reply: We appreciate reviewer's good suggestions. We have tried to prepare the larger area hydrogenated Cs₂AgBiBr₆ solar cell, however the PCE was extremely low, which may be caused by the nonuniformity of large area perovskite film. Further studies on this subject is still continuing. Thanks again for reviewer's valuable advice.

Reviewer #2 (Remarks to the Author):

Reviewer #2: In this paper, the authors report on bandgap reduction of Cs₂AgBiBr₆ double perovskite through hydrogen plasma treatment. With an appropriate duration of treatment, the bandgap was reduced to 1.61 eV from 2.14 eV. Through DFT calculations, the authors identified interstitial H as the possible cause for the observed bandgap reduction. The incorporation of interstitial hydrogen not only reduce the bandgap, but also improve carrier mobility and lifetime. As a result, the power conversion efficiency (PCE) of Cs₂AgBiBr₆ perovskite solar cells improved significantly, with the highest PCE of 6.27%. These results seem interesting. I have the following comments:

1. The PCE of 6.27% is commendable for Cs₂AgBiBr₆ PSCs, but it is not appropriate to call it "High-efficient." Perhaps the authors may use some terms like "significantly improved efficiency."

Reply: Thank for the good suggestion. The title of this manuscript has been revised as "Hydrogenated Cs₂AgBiBr₆ for Significantly Improved Efficiency of Lead-Free Inorganic Double Perovskite Solar Cell", which may give a clear description for the improvement of PCE in hydrogenated Cs₂AgBiBr₆ PSCs.

2. The authors should report the operational stability of the Cs₂AgBiBr₆ PSCs under one sun illumination, rather than just the films.

Reply: As reviewer's suggestion, the operational stability of hydrogenated Cs₂AgBiBr₆ PSCs has been measured under one sun illumination, 85 °C in N₂, one sun illumination at 85 °C and

85% relative humidity at 85 °C as shown in Fig. 3f. All discussions have been added on page 11 in manuscript.

3. The solar cells should be better measured and reported. For example, EQE curve should be provided. J-V curves measured under reverse and forward bias should be reported. The J-V curves for the hydrogenated PSCs show a V shape, suggesting all fill factors and, therefore, efficiencies could be overestimated. Authors are suggested to measure and report power outputs, i.e., measure photocurrents at photovoltages near the maximum power output.

Reply: Thanks for the critical comments. Measurements of the current density-voltage (*J-V*) curves under reverse-forward bias and external quantum efficiency (EQE) have been added in Figs. 3b and 3e (the discussion has been added on pages 10 and 11 in revised manuscript). As reviewer's suggestion, the *J-V* curves of champion hydrogenated Cs₂AgBiBr₆ (with 600 s and 1,200 s hydrogenation time) PSCs were re-tested as shown in Fig. 3a (page 10 in revised manuscript). Meanwhile, photocurrent and PCE at maximum power output were given in Fig. 3c (page 10 in revised manuscript) and Supplementary Fig. 8.

4. The determination of bandgap from absorption spectra is not convincing. There are significant below bandgap absorptions with almost the same onset for all Cs₂AgBiBr₆ including the pristine one. What is the origin of the below bandgap absorption? Is it due defects such as local disorder/lattice distortion? It seems H incorporation is just to increase the density of such defects.

Reply: Thanks for the critical comments. As reviewer's suggestion, the external quantum efficiency (EQE), photoluminescence (PL) and time-resolved photoluminescence (TRPL) of Cs₂AgBiBr₆ perovskite solar cells with different hydrogenation time (0 s, 600 s and 1,200 s) were measured in Fig. 3e, Fig. 1e and Fig. 1g, by which origin of the extended light adsorption range as compared with the bandgap of initial Cs₂AgBiBr₆ could also be deduced.

It should point out that, some groups found the doping of Fe and Cu into Cs₂AgBiBr₆ would induce the lattice distortion as well as the sub-bandgap absorption (Adv. Funct. Mater. 2021, 2109891; Adv. Funct. Mater. 2020, 30, 2005521), which is mostly ascribed to the introduction of defect states. Meanwhile, the PL intensity and lifetime of doped Cs₂AgBiBr₆

decrease correspondingly.

However, in our research, although the doping of H^{*} could introduce the crystal distortion (Fig. 4) and expand the light adsorption range (Fig. 1d), the PL lifetime and carrier mobility increase simultaneously (Figs. 1f and 1g). Furthermore, after the doping of H^{*}, the photo-response range in EQE spectrums was largely enhanced (Fig. 3e) and a new PL peak at about 760 nm was observed (Fig. 1e). All these results indicate that H^{*} doping indeed decrease the bandgap to have an enhanced light adsorption, but not introduce the defect-state generation between the valence and conductive bands.

Furthermore, by using the detection of drive-level capacitance profiling (DLCP) in Supplementary Fig. 18, the density of trap state in Cs₂AgBiBr₆ after hydrogenating from 0 s to 1,200 s keeps almost the same level of about $9.0 \times 10^{15} \text{ cm}^{-3}$ (a comparable value with other's report in Chem. Phys. Chem. 2018, 19, 1696), which further confirm that H incorporation would not increase the density of defect states.

5. Can the disorder or lattice distortion induced by H incorporation contribute to the reduced band gap?

Reply: As already pointing out in this response to question (4), in order to further understand the effect of H doping on inducing the lattice distortion in Cs₂AgBiBr₆ film, the XRD spectrum quantitative analysis and quasi *in situ* TEM characterization were performed as shown Fig. 4. Here, lattice expansion could be observed as the H insertion at the initial 0-1,200 s hydrogenation treatment. To further understand the effect of H doping on altering the bandgap of Cs₂AgBiBr₆ film, first-principle calculations were performed as shown in Fig. 5e and Supplementary Figs. 15-17, we could find that the doping H^{*} into Cs₂AgBiBr₆ lattice could only raise the energy level of VBM, which finally changes the bandgap of hydrogenated Cs₂AgBiBr₆ film.

6. What is the estimated H concentration in the hydrogenated Cs₂AgBiBr₆? Can H₂ molecules form in Cs₂AgBiBr₆?

Reply: Thanks for the critical comment. To estimate the H doping concentration in hydrogenated Cs₂AgBiBr₆ film, quantitative analysis on the XRD spectrum is performed in

Fig. 4. At the initial 1,200 s hydrogenation treatment, a low angle shift of $\text{Cs}_2\text{AgBiBr}_6$ -(044) XRD peak could be identified from 45.506° to 45.480° , which corresponds to a lattice expansion of about $\Delta a=0.006 \text{ \AA}$ (**Fig. 4c**). According to the first-principle calculation results in **Supplementary Table. 4**, such a lattice expansion corresponds to a H doping concentration less than 0.3125 at.% (1/320 atomic concentration, **page 13 in revised manuscript**).

In view of the insertion of H_2 molecule into $\text{Cs}_2\text{AgBiBr}_6$ lattice, three different positions for $\text{H}_{2-1}(\text{in})$, $\text{H}_{2-2}(\text{in})$ and $\text{H}_{2-3}(\text{in})$ were considered as shown in **Supplementary Fig. 17**. In the view of the formation energy, the three H_2 insertion configurations are all possible comparing with the interstitial H^* in the $\text{H}_n(\text{in})$ configurations. However, for the configuration of $\text{H}_{2-1}(\text{in})$ and $\text{H}_{2-3}(\text{in})$, both the energy level of CBM and VBM are with almost no change as compared with the ones in host $\text{Cs}_2\text{AgBiBr}_6$ (**Supplementary Fig. 14a**), which apparently fit not well with the decreased bandgap in **Fig. 1d**. While, for the configuration of $\text{H}_{2-2}(\text{in})$, although the bandgap seems to be reduced. However, with the reduction of CBM, the new band only improves the energy level of valence state, which is against with the theoretical and experimental values in **Fig. 2e** and **Fig. 5e**. Therefore, all these results indicate that the insertion of H_2 into the $\text{Cs}_2\text{AgBiBr}_6$ lattice is not the main contribution for the optimization of bandgap in hydrogenated $\text{Cs}_2\text{AgBiBr}_6$.

7. For TEM study, particularly for HAADF, did electron beam damage the sample?

Reply: As reviewer's comment, halide perovskite is dose sensitive under the electron beam irradiation. During the last few years, extensive efforts have tried to understand the beam damage on the structural decomposition of organic-inorganic halide perovskite, which finally disclosed that the low electron dose technique could be an effective way on capturing of the actual structure of perovskite. In this new manuscript, an *in situ* study on the beam damage on the crystal structure of $\text{Cs}_2\text{AgBiBr}_6$ was performed in **Supplementary Fig. 11**, from which we could find that a controlling of electron dose below $300 \text{ e}^-/\text{\AA}^2$ could avoid the electron beam damage on $\text{Cs}_2\text{AgBiBr}_6$ effectively. So, in our experiment, electron dose for both TEM and HAADF images were strictly controlled below $300 \text{ e}^-/\text{\AA}^2$. The electron dose rate was controlled to be $1 \text{ e}^-/\text{\AA}^2\text{s}$ for TEM model and the acquisition time for each TEM image is 1 s (**Figs. 4d-4g**, **Supplementary Fig. 11** and **Supplementary Fig. 12**). While, for the HAADF

images in Figs. 4h and 4i, the electron dose rate is about $2 \times 10^6 \text{ e}^-/\text{\AA}^2\text{s}$, and the acquisition time of each pixel is about 2 μs . We have added the experimental details on page 23 of the revised manuscript.

Reviewer #3 (Remarks to the Author):

Reviewer #1: The manuscript entitled "Hydrogenated Cs₂AgBiBr₆ for High efficient Lead free Inorganic Double Perovskite solar cell" reports a new class of materials to be used as solar cells after being treated in hydrogen plasma. The plasma treatment durations is a parameter in the study. These devices are manufactured by spin coating from a DMSO solution of the precursor compounds followed by steps of annealing and hydrogen plasma treatments. The working hypothesis of this article is that the presence of hydrogen, introduced in the structure during a plasma treatment, influences the bandgap and solar to electrical conversion efficiency of the devices. having hydrogen species with unidentified oxidation state, i.e. a radical, an ion or an atom, DFT calculations are reported.

In order to ensure that these effects can be attributed to the "hydrogen species" in the lattice, the authors must provide details about the following:

Reply: Thanks for reviewer's comments, we will reply all the suggestions one-by-one as follows.

1. XRD patterns given in the supplementary information section of the manuscript exhibit distinct formation of a new structure, identified by the authors as due to CsBr(110). Despite the authors claims that this phase forms only after 1800 s exposure, one would suspect that these species are present, but the grain size of CsBr crystallites was not large enough to be detectable by XRD at lower plasma exposure times. While the intensity of the peak due to CsBr is increasing, the intensity of the peaks that are due to the pristine perovskite crystal systematically decrease with increasing hydrogen plasma exposure period. As a result, it is strongly recommended that the authors provide the XRD patterns of the samples that were exposed to plasma for shorter periods with a thorough analysis of the peak intensities, and line widths. As far as the decomposition process is concerned starting from a Cs₂AgBiBr₆,

after the formation of CsBr, the residual should have a stoichiometry of CsAgBiBr₃. As such this is a completely different material. If CsBr is detectable in the XRD pattern after a certain hydrogen exposure period, a thorough and quantitative analysis of XRD patterns of the prepared compounds is needed. Only after the assignments of the existing structures and the impurity levels are quantified, a discussion about the presence and the role of hydrogen is meaningful. The authors are strongly encouraged to reveal the role of the new chemical environments and their structure in the light absorption properties and carrier mobilities. While it is not easy to identify new structures experimentally, DFT can provide assistance of the new compounds and their XRD patterns. The XPS data provided in Figure 4 for Cs should also be reinterpreted within the context of the CsBr compound and the CsAgBiBr₃ that should form as a result.

Reply: Thanks very much for reviewer's good comments. From the suggestions above, the most important thing is we should find out when the CsBr phase actually generate during the hydrogenation treatment of Cs₂AgBiBr₆.

In order to clarify the phase evolution of Cs₂AgBiBr₆ film during hydrogenation treatment, XRD measurement and selected area electron diffraction (SAED) of a quasi *in situ* TEM observation were performed as shown in Fig. 4 and Supplementary Fig. 10. The CsBr (110)-XRD peak was not detected until the hydrogenation treatment time reached 1,400 s (Fig. 4a). With slight reduction of peak intensity, the full width at half maximum (FWHM) of Cs₂AgBiBr₆-(044) barely change until the hydrogenation treatment exceeds 1,200 s. At this time, the low angle shift of Cs₂AgBiBr₆-(044) XRD peak from 45.506° to 45.480° indicates a lattice expansion of about $\Delta a=0.006$ Å (1200 s hydrogenation, Fig. 4c). To further confirm this phenomenon, quasi *in situ* TEM observation for the hydrogenated Cs₂AgBiBr₆ was shown in Figs. 4d-4g and Supplementary Fig. 12. Here, we could find that CsBr generation only appears after the hydrogenation treatment larger than 1,200 s, which matches well with the XRD analyses (Fig. 4a and Fig. 4b).

It should point out that, the champion PCE of hydrogenated Cs₂AgBiBr₆ PSCs presents in the 1,200 s hydrogenated samples and only the insertion of hydrogen inducing lattice expansion presents at this time. So, it is reasonable to summarize that the regulation down of bandgap and increasing of light harvest for highly efficient hydrogenated Cs₂AgBiBr₆ PSCs

are mostly ascribing to the insertion of hydrogen atom into Cs₂AgBiBr₆ lattice, rather than the decomposition intermediates.

All the comprehensive discussions have been added on pages 11 and 12 in revised manuscript.

2. The plasma treatment details are not given to see the local temperatures. If the temperatures are sufficiently high, hydrogen diffusion is going to be very rapid. Without a precise temperature distribution, it is not meaningful to extract a hydrogen diffusion coefficient based on the color profile of the films. Of course this argument is only meaningful if we have a material with intact crystal structure, a point that needs to be justified in the light of the comments made in item 1 above.

Reply: Thanks very much for the kind reminding on the experimental details of this work. Normally, plasma instruments can be divided into low temperature and high temperature plasmas. (Eur. Phys. J. D. 2016, 70, 251) In order to quantitatively analyze the temperature raising in the plasma chamber, Langmuir probe technology is firstly used as shown in Supplementary Table. 5. With the power increasing from 20 w to 40 w (working pressure at 1.4 Pa), the average electron temperature in both center and side positions are uniform and lower than 10 eV, which means the plasma treatment used in our experiment belongs to low temperature plasma.

Besides, in order to intuitively present the temperature raising during the plasma treatment, irreversible surface temperature indicating strips (ISTIS, Thermax, temperature error of ± 1 °C) were put into the plasma chamber at the working condition as shown in Supplementary Fig. 19b. However, after opening the plasma equipment (with the source power of 30 W in Ar and H₂ (80 wt.% H₂) mix gas) for 1,200 s, color of the ISTIS is unchanged (Supplementary Fig. 19c), which indicates a low temperature less than 37 °C keeping in working condition of plasma chamber. Furthermore, after the plasma treatment, the ISTIS was again put on the heating stage, the color of the specific grids would change from white to black, then the local temperature could be read directly from the ISTIS (Supplementary Figs. 19d and 19e).

To sum up, due to the application of low-temperature plasma to treat the Cs₂AgBiBr₆

samples, the temperature of sample preparation is always below 37 °C. Therefore, the effect of temperature on the diffusion of atomic hydrogen into Cs₂AgBiBr₆ lattice could be neglected reasonably.

3. The concentration profile given in equation 2 is the result of the solution of Fick's second law in cartesian coordinates in one dimension. The process described in the manuscript needs to be justified whether the cartesian representation of the diffusion problem is justifiable, i.e. sample is uniformly exposed to plasma by a steady (i.e. not transient) exposure. Only after this justification, the methodology described can be used to extract hydrogen diffusion coefficient.

Reply: Thanks very much for the good comments. The plasma is with a power output lower than 40 Watts and equipped with the radio frequency (RF) ion source of 13.56 MHz (Plasma 2021, 4, 332–344). Thus, the molecular pump in the working plasma equipment keeps the stable gas pressure, and the sample could be considered as a uniformly exposure of steady plasma environment (pages 20 and 21 in revised manuscript). Thanks again for review's valuable comments.

REVIEWER COMMENTS

Reviewer #2 (Remarks to the Author):

I thank the authors for addressing my comments very carefully and successfully. The current manuscript is acceptable for publication.

Reviewer #3 (Remarks to the Author):

The revised version of the manuscript entitled "Hydrogenated Cs₂AgBiBr₆ for High efficient Lead free Inorganic Double Perovskite solar cell" reports about improving the photon conversion efficiency of the double perovskite Cs₂AgBiBr₆ by hydrogen plasma treatment. The plasma treatment durations is a parameter investigated in the study. The parent material is prepared by spin coating from a DMSO solution followed by steps of annealing and hydrogen plasma treatments. The working hypothesis of this article is that the presence of interstitial atomic hydrogen, introduced in the structure during a plasma treatment, influences the bandgap and solar to electrical conversion efficiency of the devices.

The manuscript presents new and original information about a double perovskite system for solar energy conversion processes. The evidence for the material performance is convincing. However, the fundamental reasons of the changes induced upon hydrogen plasma exposure needs elaboration.

Exposure to hydrogen plasma induces chemical changes in these materials as evidenced by the characteristic features appearing for the CsBr compounds in the XRD patterns, after extended durations (>1200 s) of plasma treatment. The revised version of the manuscript provides a detailed structural and material characterization analysis providing further evidence on the structural transformations. The XRD data and their analysis indeed reveal structural changes. However, the chemical transformation that induces formation of CsBr as reported by the authors needs further refinement which should include the potential alternative molecules that can be present after the plasma treatment leading to the formation of CsBr. It is highly probable that the material with the superior properties after hydrogen plasma treatment may have impurity phases other than CsBr. It is also possible that these impurities are also responsible for the reported superior properties. Along these lines, the authors are encouraged to look for the residues of HBr in the plasma chamber which is a very highly likely candidate product of the plasma treatment process of the parent compound. As a result of the decomposition leading to the formation of the CsBr (and perhaps HBr) formation of other compounds from the decomposition of parent Cs₂AgBiBr₆ will be highly likely. A discussion inviting for further study about the compounds that may form as a result of the hydrogen plasma treatment is needed for the thoroughness of the discussion.

In response to a concern raised for the previous submission about the local temperatures during the plasma treatment, the authors have reported a thorough experimental study and local temperature measurements. Additional data, and the discussion for the temperature measurement is indeed convincing about the bulk temperature of the plasma chamber. However, one would not expect thermal equilibrium at the specific site at the atomic level where highly energetic hydrogen plasma interacts with the solid material. The surface temperature indicating strips are highly likely to reflect the thermal steady state temperature of the plasma chamber with its environment, but not the temperature of the chemical events at the atomic scale. This concern is raised to emphasize the need to consider the local chemistry possible at higher temperatures, that may lead to chemical transformations leading to the formation of CsBr and other compounds.

The main article is missing the time dependent diffusion equation that the authors used to estimate the process (equation 2).

Figure 4 (part c) the x axis label needs to be changed from 2 Theta to "Hydrogenation duration"

Elaboration on the language is strongly encouraged.

Deniz Uner
Chemical Engineering
Middle East Technical University
Ankara 06800 Turkey

Reviewer#3 (Remarks to the Author)

1. *The manuscript presents new and original information about a double perovskite system for solar energy conversion processes. The evidence for the material performance is convincing. However, the fundamental reasons of the changes induced upon hydrogen plasma exposure needs elaboration.*

Exposure to hydrogen plasma induces chemical changes in these materials as evidenced by the characteristic features appearing for the CsBr compounds in the XRD patterns, after extended durations (>1,200 s) of plasma treatment. The revised version of the manuscript provides a detailed structural and material characterization analysis providing further evidence on the structural transformations. The XRD data and their analysis indeed reveal structural changes. However, the chemical transformation that induces formation of CsBr as reported by the authors needs further refinement which should include the potential alternative molecules that can be present after the plasma treatment leading to the formation of CsBr. It is highly probable that the material with the superior properties after hydrogen plasma treatment may have impurity phases other than CsBr. It is also possible that these impurities are also responsible for the reported superior properties. Along these lines, the authors are encouraged to look for the residues of HBr in the plasma chamber which is a very highly likely candidate product of the plasma treatment process of the parent compound. As a result of the decomposition leading to the formation of the CsBr (and perhaps HBr) formation of other compounds from the decomposition of parent Cs₂AgBiBr₆ will be highly likely. A discussion inviting for further study about the compounds that may form as a result of the hydrogen plasma treatment is needed for the thoroughness of the discussion.

Reply: Thanks for the good suggestion. As shown in previous XRD and TEM measurements in Fig. 4, the hydrogenated Cs₂AgBiBr₆ film would not proceed a decomposition process into CsBr under 1,200 s hydrogenation treatment, and the hydrogenated Cs₂AgBiBr₆ solar cells exhibit a record high photoelectric conversion efficiency (PCE) at this time (Fig. 3). In this comment, the reviewer cares about the decomposition products such as CsBr, which only appears after the hydrogenation time excess 1,200 s.

In order to further understand the decomposition pathway of Cs₂AgBiBr₆ film, the gas production in plasma chamber was collected when enlarging the hydrogenation time up to

2,400 s (Supplementary Fig. 13d and Methods). After detecting the chemical composition of this gas production, a peak at 12.46 min appears in the ion chromatography spectrum with a total area of 0.007 $\mu\text{S}/\text{cm}$ (Supplementary Fig. 13d). As comparing with the standard samples of HBr gas with a concentration of 10 ppm and 0.01 ppm, the peak of ion chromatography spectrum at 12.36-12.52 min could be identified as the characteristic peak of Br^- deriving from HBr (Supplementary Figs. 13a and 13b). Therefore, after hydrogenating $\text{Cs}_2\text{AgBiBr}_6$ film in plasma up to 2,400 s, the film has been decomposed into HBr gas.

In order to check the solid-state decomposition production after long-term hydrogenation treatment, the structural analyses were performed previously in Fig. 4 and Supplementary Fig. 10, from which a new XRD peak at 29.375° presented (Fig. 4a) and could be identified as CsBr (110) and possibly overlapping BiBr ($\bar{3}12$). By using quasi *in situ* TEM observation (Figs. 4d-4g), the CsBr (100) was evidently confirmed. Considering the decomposition products above, the decomposition pathway of $\text{Cs}_2\text{AgBiBr}_6$ under long-term hydrogenation treatment could be possibly written as: $\text{Cs}_2\text{AgBiBr}_6 + 2\text{H} \rightarrow 2\text{CsBr} + 2\text{HBr} + \text{BiBr} + \text{AgBr}$. However, due to the metastable property of AgBr and the weak XRD peak intensity contribution, the existence of AgBr during the decomposition of $\text{Cs}_2\text{AgBiBr}_6$ is hard to be checked in the spectrum (Supplementary Fig. 10).

It should point out that, as collecting and detecting the gas production after a hydrogenation treatment less than 1,200 s (Supplementary Fig. 13c), almost no HBr peak could be recognized at 12.36-12.52 min, which proves that no $\text{Cs}_2\text{AgBiBr}_6$ has been decomposed, and the PCE improvement of hydrogenated $\text{Cs}_2\text{AgBiBr}_6$ perovskite solar cell is only induced by the insertion of atomic hydrogenation into the $\text{Cs}_2\text{AgBiBr}_6$ crystal lattice.

2. In response to a concern raised for the previous submission about the local temperatures during the plasma treatment, the authors have reported a thorough experimental study and local temperature measurements. Additional data, and the discussion for the temperature measurement is indeed convincing about the bulk temperature of the plasma chamber. However, one would not expect thermal equilibrium at the specific site at the atomic level where highly energetic hydrogen plasma interacts with the solid material. The surface

temperature indicating strips are highly likely to reflect the thermal steady state temperature of the plasma chamber with its environment, but not the temperature of the chemical events at the atomic scale. This concern is raised to emphasize the need to consider the local chemistry possible at higher temperatures, that may lead to chemical transformations leading to the formation of CsBr and other compounds.

Reply: Thanks for the good suggestion. We have estimated the temperature of the chemical events at the atomic scale and added the following discussion in the part of “Plasma treatment on the perovskite film” of “Methods” section (see Pages 21-22)

In addition to the environment temperature of the thermal steady state of the plasma chamber, the temperature of the chemical events at the atomic scale should also be taken into account. To describe the chemical interaction between plasma gas and Cs₂AgBiBr₆ film, the maximum energy transfer T_{\max} from highly energetic hydrogen radicals to the surface atom of Cs₂AgBiBr₆ is considered⁵⁶,

$$T_{\max} = \frac{2ME(E+2mc^2)}{(m+M)^2c^2+2ME} \quad (3)$$

here, E is kinetic energy of the incoming plasma excited hydrogen radical; m , M is the mass of hydrogen atom and the chemical elements in Cs₂AgBiBr₆; c is the velocity of light (3.00×10^8 m/s). If the kinetic energy E is much smaller than mc^2 ($mc^2 = 9.40 \times 10^8$ eV for the hydrogen atom), the equation could be simplified as

$$T_{\max} = E \frac{4Mm}{(m+M)^2} \quad (4)$$

In this work, the maximum kinetic energy of radicals in hydrogen plasma is 9.3 eV (Supplementary Table 5), which is much smaller than mc^2 . So, the energy transfer from the hydrogen radicals to the surface atoms in Cs₂AgBiBr₆ could be deduced from equation 4. Based on this, the maximum energy transfer T_{\max} to the surface atoms of Cs, Ag, Bi and Br could be determined as 0.28 eV, 0.34 eV, 0.18 eV and 0.46 eV, respectively, which are obviously lower than the bond dissociation energies of Cs-Br (4.03 eV), Bi-Br (2.49 eV) and Ag-Br (2.91 eV) in Cs₂AgBiBr₆ (Supplementary Table 6)⁵⁷. Therefore, the calculation results indicate that the low temperature hydrogen plasma treatment is hard to break the chemical bonds in Cs₂AgBiBr₆.

However, the Cs₂AgBiBr₆ lattice is stable only when the hydrogenation treatment is less than

1,200 s. The decomposition of hydrogenated Cs₂AgBiBr₆ phase occurs after 1,400 s hydrogenation treatment (Fig. 4), which is attributed to the intrinsic structural instability of hydrogenated Cs₂AgBiBr₆. As increasing the concentration of atomic hydrogen into Cs₂AgBiBr₆ lattice, the formation energy of perovskite does process a raising (Fig. 5d).

3. The main article is missing the time dependent diffusion equation that the authors used to estimate the process (equation 2).

Reply: In this work, the hydrogen plasma was operated at a constant gas pressure, the diffusion source concentration of H* at the surface of Cs₂AgBiBr₆ could be identified as time independent. So, the diffusion profile of H* in Cs₂AgBiBr₆ film should obey the constant surface concentration diffusion model of equation 2. Furthermore, the diffusion equation 2 with diffusion time (different hydrogenation time of 600 s, 1,200 s, 1,800 s, 2,400 s and 3,000 s) has been considered in Supplementary Fig. 4 and “Method” section on pages 18 and 19. As comparing with the experimental data in Fig. 2a, the diffusion coefficients of hydrogen atom in Cs₂AgBiBr₆ material is determined to be in the range of 1×10^{-14} cm²/s to 1×10^{-13} cm²/s, from which the concentration distribution of atomic hydrogen in 1,200 s hydrogenated Cs₂AgBiBr₆ film has been deduced in Fig. 2b.

4. Figure 4 (part c) the x axis label needs to be changed from 2 Theta to “Hydrogenation duration”

Reply: Thanks for pointing out the mistake in Fig. 4c. As reviewer’s comment, the x axis label has been changed from “2 Theta” to “Hydrogenation time”.

5. Elaboration on the language is strongly encouraged.

Reply: Thanks for the reviewer’s good suggestion. The language of the manuscript has been carefully improved.

REVIEWERS' COMMENTS

Reviewer #3 (Remarks to the Author):

The manuscript entitled "Hydrogenated Cs₂AgBiBr₆ for Significantly Improved Efficiency of Lead-Free Inorganic Double Perovskite Solar Cell" reports carefully conducted research on the role of the plasma hydrogenation on the PV performance of a double perovskite Cs₂AgBiBr₆. The present revisions and additional data satisfactorily address all of the concerns raised by this reviewer.

The following minor typing corrections are brought to the attention of the authors:

1. Frequent use of "It should point out" needs to be replaced with "it should be pointed out"
2. Line 16 greatly needs to be replaced with great
3. Line 76 Deniz Uner et al needs to be replaced with Mete et al. This is a kind gesture by the authors. Thank you.
4. Line 380 The sentence needs clarification. "...relatively high value of ..."
5. Line 465 The equation has a typing error, resulting in a dimensional inconsistency. Please check!

regards;

Deniz Uner
Professor of Chemical Engineering
Middle East Technical University
Ankara 06800 TURKEY

Point to Point Response

Reviewer#3 (Remarks to the Author)

The following minor typing corrections are brought to the attention of the authors:

1. Frequent use of "It should point out" needs to be replaced with "it should be pointed out"

Reply: Thanks for the suggestion. The sentences of "It should point out" in the manuscript have been replaced by "it should be pointed out".

2. Line 16 greatly needs to be replaced with great.

Reply: The word "greatly" has been replaced with "great" on page 1, thanks very much.

3. Line 76 Deniz Uner et al needs to be replaced with Mete et al. This is a kind gesture by the authors. Thank you.

Reply: Thanks again. The name of first author "Mete et al" in reference 32 has been revised in new revision on page 4.

4. Line 380 The sentence needs clarification. "...relatively high value of ..."

Reply: In order to avoid the misunderstanding of the sentence "...relatively high value of ...", here we revised it into "As we all known, the solubility of $\text{Cs}_2\text{AgBiBr}_6$ in dimethylsulfoxide (DMSO) solution is about 0.6 mol/L, here we choose DMSO as the precursor solvent" on page 15.

5. Line 465 The equation has a typing error, resulting in a dimensional inconsistency. Please check!

Reply: As reviewer's suggestion, the typing error " T_{\max} " in equations (3) and (4) have been modified as " E_T " on page 22. Thanks very much for the advises.

Formatting correction

1. Atomic orbital notations and corresponding XPS labels have been modified as italics format.

2. Abbreviations were added when they were firstly appeared in main text.

3. The section of "Results" has been divided into subsections with a title within 60 characters.

4. The description about single points, median, 25th, 75th percentile, maximum and minimum in box plots were added in figure legends (Figure 3d) on page 35 in manuscript and

(Supplementary Figure 9) on page 9 in supplementary information.

5. We added a brief title for Figure 5 on page 36.

6. The function of “erfc” in equation 2 on page 8 has been revised with a format in roman.

7. Supplementary Tables 1-6 in supplementary information were modified from non-editable format to editable format.